# `Neur2SP`: Neural Two-Stage Stochastic Programming

**Justin Dumouchelle**[*]     **Rahul Patel**[*]     **Elias B. Khalil**[†]     **Merve Bodur**
Department of Mechanical & Industrial Engineering, University of Toronto

## Abstract

Stochastic Programming is a powerful modeling framework for decision-making under uncertainty. In this work, we tackle two-stage stochastic programs (2SPs), the most widely used class of stochastic programming models. Solving 2SPs exactly requires optimizing over an expected value function that is computationally intractable. Having a mixed-integer linear program (MIP) or a nonlinear program (NLP) in the second stage further aggravates the intractability, even when specialized algorithms that exploit problem structure are employed. Finding high-quality (first-stage) solutions – without leveraging problem structure – can be crucial in such settings. We develop `Neur2SP`, a new method that approximates the expected value function via a neural network to obtain a surrogate model that can be solved more efficiently than the traditional extensive formulation approach. `Neur2SP` makes no assumptions about the problem structure, in particular about the second-stage problem, and can be implemented using an off-the-shelf MIP solver. Our extensive computational experiments on four benchmark 2SP problem classes with different structures (containing MIP and NLP second-stage problems) demonstrate the efficiency (time) and efficacy (solution quality) of `Neur2SP`. In under 1.66 seconds, `Neur2SP` finds high-quality solutions across all problems even as the number of scenarios increases, an ideal property that is difficult to have for traditional 2SP solution techniques. Namely, the most generic baseline method typically requires minutes to hours to find solutions of comparable quality.

## 1   Introduction

Mathematical programming consists of a gamut of tools to solve optimization problems. Under perfect information, i.e., when all the data is deterministic and known, many of these problems can be solved as a linear program (LP) or mixed-integer linear program (MIP). However, in many cases, there is a need to deal with problems with partial information. Stochastic programming is one such framework that allows us to incorporate uncertainty into decision-making.

In this work, we focus our attention on Two-stage Stochastic Programs (2SPs). A 2SP involves two sets of decisions, namely the first-stage and second-stage (recourse) decisions, to be made before and after the uncertainty is realized, respectively. Given the (joint) probability distribution of the random parameters of the problem, the most common objective of 2SP is to optimize the expected value of the decisions. For example, in a two-stage stochastic facility location problem, first-stage decisions consist of which facilities should be built whereas second-stage decisions involve assigning customers to open facilities to meet their stochastic demand, and the overall objective is to minimize the sum of the cost of the first-stage decisions and the expected cost of the second-stage decisions.

2SPs are usually solved via Sample Average Approximation (SAA), which limits the future uncertainty to a finite set of possible realizations (scenarios). The SAA approximation of a 2SP is a

---

[*]These authors contributed equally.

[†]Corresponding author: `khalil@mie.utoronto.ca`.

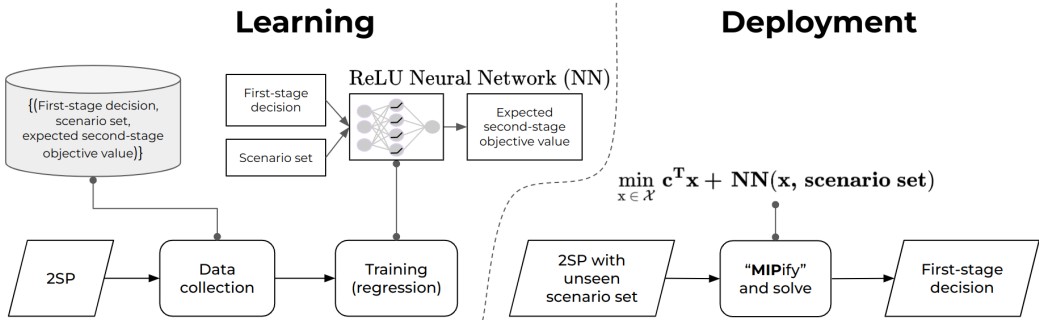

Figure 1: Overview of `Neur2SP`. The leftmost block is the input, namely, a 2SP. From the 2SP, we follow the data generation procedure from Section 4.3 to obtain a dataset consisting of tuples of (first-stage decision, scenario set, corresponding expected second-stage objective value). We then train one of the learning models presented in Section 4.1 to predict the expected cost given a first-stage decision and scenario set. The trained model is then embedded into a MIP using the procedure in Section 4.2 to obtain an approximate MIP (the "MIPify" step). Lastly, the approximate MIP is solved with an off-the-shelf MIP solver to obtain a first-stage 2SP solution.

reduction to an equivalent deterministic problem and can be solved by the so-called *extensive form*: a monolithic formulation where scenario copies of the second-stage decision variables are created and linked to the first-stage decisions. However, even for small 2SPs, solving the extensive form may be intractable as it requires introducing a large number of (possibly integer) variables and (possibly nonlinear) constraints. As such, specialized algorithms are required. If the second-stage problem assumes the form of an LP, then algorithms such as Benders' decomposition (also known as the L-shaped method) can be leveraged to efficiently solve the problem to optimality. Unfortunately, in many practical applications of 2SP, the second-stage problem assumes the form of a MIP, for which specialized decomposition algorithms might not be efficient. The existence of continuous first-stage variables linked to the second-stage problem significantly increases the difficulty of solving such problems. This is exacerbated when the second-stage problem is nonlinear, for which no general and structure-agnostic solution strategy exists.

In this work, we propose `Neur2SP`, a framework for constructing an easier-to-solve surrogate optimization problem for 2SP with the use of supervised deep learning. In a nutshell, a Rectified Linear Unit (ReLU) neural network is trained to approximate the second-stage objective value for a set of scenarios. Using MIP-representable activation functions such as the ReLU, the forward computation of the trained network can be embedded into a MIP. The surrogate problem is then confined to optimizing *only* first-stage decisions with respect to the first-stage objective function and the neural network approximation of the second-stage objective [3]. Assuming a small and accurate neural network can be used, the surrogate problem is much smaller than the extensive form, and thus faster to solve. The entire procedure is summarized in Figure 1. Our main contributions are as follows:

1. **Novelty:** `Neur2SP` is the first generic machine learning approach for deriving a heuristic solution for 2SP. We introduce a highly parallelizable data collection procedure and show two separate neural models which can be used to formulate a deterministic mixed-integer surrogate problem for 2SP;

2. **Generality:** `Neur2SP` can be used out-of-the-box for 2SPs with linear and nonlinear objectives and constraints as well as mixed-integer variables in both the first and second stages, all without using any problem structure, i.e., in a purely data-driven way;

3. **Performance:** `Neur2SP` is shown to produce high-quality solutions significantly faster than the solely applicable general baseline method, the extensive form approach, for a variety of benchmark problems, namely, stochastic facility location problem, an investment problem, a server location problem, and a pooling problem from chemical engineering.

---

[3]For a fixed first-stage solution obtained via this surrogate, an optimal second-stage decision can be obtained relatively quickly for each scenario if desired.

## 2 Preliminaries

We introduce the 2SP setting and describe the MIP formulation for a ReLU activation function which is central to the surrogate model we propose in this work. Appendix A summarizes the notation used.

### 2.1 Two-stage Stochastic Programming

A 2SP can be generally expressed as $\min_{\mathbf{x}}\{\mathbf{c}^\mathsf{T}\mathbf{x} + \mathbb{E}_\xi[Q(\mathbf{x},\boldsymbol{\xi})] : \mathbf{x} \in \mathcal{X}\}$, where $\mathbf{c} \in \mathbb{R}^n$ is the first-stage cost vector, $\mathbf{x} \in \mathbb{R}^n$ represents the first-stage decisions, $\mathcal{X}$ is the first-stage feasible set, and $\boldsymbol{\xi}$ is the vector of random parameters that follow a probability distribution $\mathbb{P}$ with support $\Xi$. The *value function* $Q : \mathcal{X} \times \Xi \to \mathbb{R}$ returns the cost of optimal second-stage (recourse) decisions under realization $\boldsymbol{\xi}$ given the first-stage decisions of $\mathbf{x}$. In many cases, as the $Q(\mathbf{x},\boldsymbol{\xi})$ is obtained by solving a mathematical program, evaluating the *expected value function* $\mathbb{E}_\xi[Q(\mathbf{x},\boldsymbol{\xi})]$ is intractable.

To provide a more tractable formulation, the extensive form (EF) is used. Using a set of $K$ scenarios, $\boldsymbol{\xi}_1, \ldots, \boldsymbol{\xi}_K$, sampled from the probability distribution $\mathbb{P}$, $\text{EF}(\boldsymbol{\xi}_1, \ldots, \boldsymbol{\xi}_K) \equiv \min_{\mathbf{x}}\{\mathbf{c}^\mathsf{T}\mathbf{x} + \sum_{k=1}^{K} p_k Q(\mathbf{x},\boldsymbol{\xi}_k) : \mathbf{x} \in \mathcal{X}\}$, where $p_k$ is the probability of scenario $\boldsymbol{\xi}_k$ being realized. If $Q(\mathbf{x},\boldsymbol{\xi}) = \min_{\mathbf{y}}\{F(\mathbf{y},\boldsymbol{\xi}) : \mathbf{y} \in \mathcal{Y}(\mathbf{x},\boldsymbol{\xi})\}$, then $\text{EF}(\boldsymbol{\xi}_1, \ldots, \boldsymbol{\xi}_K)$ can be expressed as

$$\min_{\mathbf{x},\mathbf{y}} \left\{ \mathbf{c}^\mathsf{T}\mathbf{x} + \sum_{k=1}^{K} p_k F(\mathbf{y}_k,\boldsymbol{\xi}_k) : \mathbf{x} \in \mathcal{X}, \mathbf{y}_k \in \mathcal{Y}(\mathbf{x},\boldsymbol{\xi}_k) \, \forall k = 1, \ldots, K \right\},$$

which can be solved through standard deterministic optimization techniques. However, the number of variables and constraints of the EF grows linearly with the number of scenarios. Furthermore, if $Q(\cdot,\cdot)$ is the optimal value of a MIP or an nonlinear program (NLP), the EF model becomes significantly more challenging to solve as compared to the LP case, limiting its applicability even at small scale.

### 2.2 Embedding Neural Networks into MIPs

Mathematically, an $\ell$-layer fully-connected neural network can be expressed as: $\mathbf{h}^1 = \sigma(W^0\boldsymbol{\alpha} + \mathbf{b}^0)$; $\mathbf{h}^{m+1} = \sigma(W^m\mathbf{h}^m + \mathbf{b}^m), m = 1, \ldots, \ell - 1; \beta = W^\ell\mathbf{h}^\ell + \mathbf{b}^\ell$. Here, $\boldsymbol{\alpha} \in \mathbb{R}^m$ is the input, $\beta \in \mathbb{R}$ is the prediction, $\mathbf{h}^i \in \mathbb{R}^{d_i}$ is the $i$-th hidden layer, $W^i \in \mathbb{R}^{d_i \times d_{i+1}}$ is the matrix of weights from layer $i$ to $i + 1$, $\mathbf{b}^i \in \mathbb{R}^{d_i}$ is the bias at the $i$-th layer, and $\sigma$ is a non-linear activation function, here the activation function is given by $\text{ReLU}(a) = \max\{0, a\}$ for $a \in \mathbb{R}$.

Central to Neur2SP is the embedding of a trained neural network into a MIP. Here, we present the formulation proposed by [Fischetti and Jo, 2018]. For a given hidden layer $m$, the $j$-th hidden unit, $h_j^m$, can be written as

$$h_j^m = \text{ReLU}\left(\sum_{i=1}^{d_{m-1}} w_{ij}^{m-1} h_i^{m-1} + b_j^{m-1}\right), \tag{1}$$

where $w_{ij}^m$ is the element at the $j$-th row and $i$-th column of $W^{m-1}$ and $b_j^{m-1}$ is the $j$-th index of $\mathbf{b}^{m-1}$. To model ReLU in a MIP for the $j$-th unit in the $m$-th layer, we use the variables $\hat{h}_j^m, \check{h}_j^m$ and $\hat{h}_i^{m-1}$ for $i = 1, \ldots, d_{m-1}$. The ReLU activation is then modeled with the following constraints:

$$\sum_{i=1}^{d_{m-1}} w_{ij}^{m-1}\hat{h}_i^{m-1} + b_j^{m-1} = \hat{h}_j^m - \check{h}_j^m, \tag{2a}$$

$$z_j^m = 1 \Rightarrow \hat{h}_j^m \leq 0, \tag{2b}$$

$$z_j^m = 0 \Rightarrow \check{h}_j^m \leq 0, \tag{2c}$$

$$\hat{h}_j^m, \check{h}_j^m \geq 0, \tag{2d}$$

$$z_j^m \in \{0, 1\}, \tag{2e}$$

where the logical constraints in Equation (2b) and Equation (2c) are translated into big-M constraints by MIP solvers. To verify the correctness of this formulation, observe that constraints (2b) and (2c) in conjunction with the fact the binary $z_j^m$ ensures that at most one of $\hat{h}_j^m$ and $\check{h}_j^m$ are non-zero.

Furthermore, since both $\hat{h}_j^m$ and $\check{h}_j^m$ are non-negative, if $\sum_{i=1}^{d_{m-1}} w_{ij}^{m-1} \hat{h}_i^{m-1} + b_j^{m-1} > 0$, then it follows that $\hat{h}_j^m > 0$ and $\check{h}_j^m = 0$. If negative, then $\hat{h}_j^m = 0$ and $\check{h}_j^m > 0$. Thus, we have that if the left-hand side of (2a) is positive, $\hat{h}_j^m$ will be positive; if it is negative, then $\hat{h}_j^m = 0$; this is an exact representation of the ReLU function.

## 3  Related Work

### 3.1  Machine Learning for Nested Optimization

Machine learning has recently been employed to solve nested optimization problems; by "nested", we mean optimization problems whose objective or constraints involve another optimization. For example, Nair et al. [2018], Shen et al. [2021], Jiang et al. [2021], Xiong and Hsieh [2020], Shao et al. [2022] directly predict a binary or continuous solution vector. The major limitation with predicting solutions directly is the inability to handle variable integrality and hard constraints. In addition, only Nair et al. [2018] consider 2SP, whereas the others focus on bi-level problems with a single inner optimization, rather than the expectation as in stochastic programming. For stochastic programming, there has been a significant interest in the integration of learning to enhance prevalent solution techniques. We specifically discuss three areas of related work: learning-enabled optimization, learning-based algorithms for stochastic programming, and scenario reduction for stochastic programming.

The line of work on learning-enabled optimization [Deng and Sen, 2022, Liu et al., 2022, Diao and Sen, 2020] introduced "predictive stochastic programming" to leverage contextual information when formulating SP models. This is in contrast to our approach, which leverages predictions to reduce computing times in a non-contextual 2SP setting. That being said, Neur2SP admits extensions to the contextual setting by including the context information during training.

In recent years, several studies have explored the use of integrating predictions within stochastic programming algorithms for computational improvements. Donti et al. [2017] proposed an end-to-end approach to directly optimize a task-loss for contextual stochastic programming problems by differentiating through the argmin operator, specifically for strongly convex problems. Dai et al. [2022] developed a model to solve multi-stage linear optimization problems by learning the piecewise value function of the nested problems. Larsen et al. [2022] leveraged predictions to improve an exact decomposition-based algorithm for 2SP. Neur2SP differs from these approaches as it can be applied to problems with both hard constraints and integer/non-linear second-stage problems.

Lastly, another related research direction for learning-based stochastic programming is scenario reduction, which reduces the complexity of the stochastic programming problem by finding a smaller set of "representative scenarios". Many of these approaches [Dupačová et al., 2003, Römisch, 2009, Beraldi and Bruni, 2014, Prochazka and Wallace, 2020, Keutchayan et al., 2021] perform some form of clustering to reduce the number of scenarios and then solve a smaller surrogate problem with these scenarios. Recently, Wu et al. [2022] used a conditional variational autoencoder to learn scenario embeddings and perform clustering on them for scenario reduction. To find representative scenarios, they use semi-supervised learning with respect to the second-stage cost. However, these predictions are not leveraged explicitly in the optimization as is done with Neur2SP. Bengio et al. [2020] predicts a representative scenario for an input scenario set and use it to form a smaller surrogate problem. They show that using the predicted representative scenario, a near-optimal first-stage decision can be obtained by solving the surrogate. However, their method requires some domain expertise as it relies on the problem structure to build the representative scenario for training.

### 3.2  Neural Network Embeddings

Neur2SP can be broadly classified as both a learning-based scenario reduction approach and a learning-accelerated heuristic for stochastic programming. The reason for this is that Neur2SP reduces the computational complexity introduced by the scenarios by computing a compact representation that is then leveraged to formulate an approximation to the EF. We specifically leverage the recent line of work by Cheng et al. [2017], Tjeng et al. [2017], Fischetti and Jo [2018], Serra et al. [2018], which studies the problem of embedding a trained neural network with ReLU activation into a MIP. The works of Anderson et al. [2020] and Grimstad and Andersson [2019] present MIP encoding

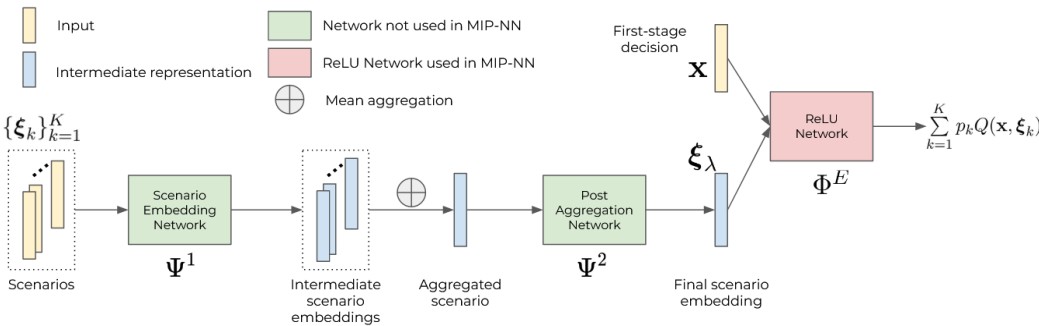

Figure 2: NN-E architecture diagram.

formulations with tighter LP relaxations by appropriately setting the big-$M$ constraints, leading to reduced solving time. The growing interest in embedding predictive models in MIPs has led to the development of libraries such as JANOS [Bergman et al., 2022] and OMLT [Ceccon et al., 2022]. Say et al. [2017], Grimstad and Andersson [2019], Murzakhanov et al. [2020], Katz et al. [2020], Kody et al. [2022] propose the use of embedded neural networks to formulate surrogate MIPs for intractable and non-linear constraints in optimization problems. To the best of our knowledge, Neur2SP is the first approach that employs this technique in stochastic programming or more generally for the simplification of nested optimization problems.

## 4 The Neur2SP Framework

In this section, we present two neural architectures, the corresponding surrogate problems that approximate a given 2SP, and a data collection strategy. Figure 1 summarizes the Neur2SP framework.

### 4.1 Neural Network Architectures

We propose two distinct neural architectures for predicting the second-stage costs: NN-E approximates the expected value of the second-stage cost of *a set of scenarios*, whereas NN-P approximates the *per-scenario* value of the second-stage cost for *a single scenario*.

**NN-E** (Figure 2) learns a mapping from $\left(\mathbf{x}, \{\boldsymbol{\xi}_k\}_{k=1}^K\right) \to \sum_{k=1}^K p_k Q(\mathbf{x}, \boldsymbol{\xi}_k)$. In words, the model takes in a first-stage solution $\mathbf{x}$ and any finite set of scenarios sampled from $\Xi$, and outputs a prediction of the expected second-stage objective value. We embed the scenario set $\{\boldsymbol{\xi}_k\}_{k=1}^K$ into a latent space by passing each scenario, independently, through the same neural network $\Psi^1$, then performing mean-aggregation over the resulting $K$ embeddings. The aggregated embedding is passed through another network, $\Psi^2$, to obtain the final embedding of the scenario set, $\xi_\lambda$. This embedding, representing the scenario set à-la-DeepSets [Zaheer et al., 2017], is appended to the input first-stage decision and passed through a ReLU feed-forward network $\Phi^E$ to predict the expected second-stage value. Hence, the final output is such that $\Phi^E(\mathbf{x}, \Psi^2(\oplus_{k=1}^K \Psi^1(p_k, \boldsymbol{\xi}_k))) \approx \sum_{k=1}^K p_k Q(\mathbf{x}, \boldsymbol{\xi}_k)$. Note that the embedding networks, $\Psi^1$ and $\Psi^2$, can be arbitrarily complex as only the latent representation is embedded into the approximate MIP. Also, although $\Psi^1$ is trained using $K$ scenarios, once the networks are trained, they can be used with any (potentially much larger) finite number of scenarios.

**NN-P** learns a mapping $\Phi^P$ from $(\mathbf{x}, \boldsymbol{\xi}) \to Q(\mathbf{x}, \boldsymbol{\xi})$ for $\boldsymbol{\xi}$ sampled from $\Xi$. Once the mapping $\Phi^P$ is learned, we can approximate the expected second-stage objective value for any finite set of scenarios as $\sum_{k=1}^K p_k Q(\mathbf{x}, \boldsymbol{\xi}_k) \approx \sum_{k=1}^K p_k \Phi^P(\mathbf{x}, \boldsymbol{\xi}_k)$. $\Phi^P$ is a feed-forward neural network with input given by the concatenation of $\mathbf{x}$ and $\boldsymbol{\xi}$.

### 4.2 Neural Network Embedding for 2SP

We now describe the surrogate MIP for both the NN-E and NN-P learning models from the preceding section. Let $\Lambda$ represent the number of predictions made by the neural network. For the NN-E case,

$\Lambda = 1$ as we only predict the expected second-stage value for a set of scenarios. In the NN-P case, $\Lambda = K$ as we predict the second-stage value for each scenario. In this section, we use $[M]$ to denote $\{1, \ldots, M\}$ for $M \in \mathbb{Z}_+$.

Let $\hat{h}_j^{m,\lambda}$ represent the ReLU output for the $j$-th hidden unit in the $m$-th hidden layer for output $\lambda$, for all $m \in [\ell - 1]$, $j \in [d_m]$, and $\lambda \in [\Lambda]$. Suppose $\check{h}_j^{m,\lambda}$ is a slack variable used to model the $ReLU$ output for the $j$-th hidden unit in the $m$-th hidden layer for scenario $k$, for all $m \in [\ell - 1]$, $j \in [d_m]$, and $\lambda \in [\Lambda]$. Let $z_j^{m,\lambda}$ be a binary variable used to ensure that at most one of $\hat{h}_j^{m,k}$ and $\check{h}_j^{m,k}$ are non-zero. This variable is defined for all $m \in [\ell - 1]$, $j \in [d_m]$, and $\lambda \in [\Lambda]$. Suppose $\beta_\lambda$ is the $\lambda$-th prediction by the neural network, for all $\lambda \in [\Lambda]$.

With the above variables we can define an approximation to EF as given in Equation (3). The objective function (3a) minimizes the sum of the cost of the first-stage decisions and the approximate cost of the second-stage value. Constraints (3b)-(3d) propagate a first-stage solution $\mathbf{x}$ to the output of the neural network for each scenario. Constraints (3e)-(3h) ensure the prediction of the neural network is respected. Constraint (3i) ensures the feasibility of the first-stage solution.

In this approximation, we introduce a number of additional variables and big-M constraints. Specifically, for a neural network with $H$ hidden units, we introduce $\Lambda \cdot H$ additional binary variables for $z_j^{m,\lambda}$. In addition, we introduce $2 \cdot \Lambda \cdot H$ continuous variables for $\hat{h}_j^{m,\lambda}$ and $\check{h}_j^{m,\lambda}$. Lastly, we require an additional $\Lambda$ variables for the output of the network. Although the number of variables we introduce in this approximation is quite large, we hypothesize that the resulting MIP will be easier to solve than the extensive form, in particular, when the second-stage problem is nonlinear. In the remainder of the paper, we refer to the surrogate MIP given in (3) as MIP-NN.

$$
\min \quad \mathbf{c}^\mathsf{T}\mathbf{x} + \sum_{\lambda=1}^{\Lambda} p_\lambda \beta_\lambda \tag{3a}
$$

$$
\text{s.t.} \quad \sum_{i=1}^{d_0} w_{ij}^0 [\mathbf{x}, \boldsymbol{\xi}_\lambda]_i + b_j^0 = \hat{h}_j^{1,\lambda} - \check{h}_j^{1,\lambda} \qquad \forall\, j \in [d_1], \lambda \in [\Lambda], \tag{3b}
$$

$$
\sum_{i=1}^{d_{m-1}} w_{ij}^{m-1} \hat{h}_i^{m-1,\lambda} + b_j^{m-1} = \hat{h}_j^{m,\lambda} - \check{h}_j^{m,\lambda} \qquad \forall\, m \in [\ell-1], j \in [d_m], \lambda \in [\Lambda], \tag{3c}
$$

$$
\sum_{i=1}^{d_\ell} w_{ij}^\ell \hat{h}_i^{\ell,\lambda} + b^\ell \le \beta_\lambda \qquad \forall \lambda \in [\Lambda], \tag{3d}
$$

$$
z_j^{m,\lambda} = 1 \Rightarrow \hat{h}_j^{m,\lambda} = 0 \qquad \forall\, m \in [\ell-1], j \in [d_m], \lambda \in [\Lambda], \tag{3e}
$$

$$
z_j^{m,\lambda} = 0 \Rightarrow \check{h}_j^{m,\lambda} = 0 \qquad \forall\, m \in [\ell-1], j \in [d_m], \lambda \in [\Lambda], \tag{3f}
$$

$$
z_j^{m,\lambda} \in \{0,1\} \qquad \forall\, m \in [\ell-1], j \in [d_m], \lambda \in [\Lambda], \tag{3g}
$$

$$
\hat{h}_j^{m,\lambda}, \check{h}_j^{m,\lambda} \ge 0 \qquad \forall\, m \in [\ell-1], j \in [d_m], \lambda \in [\Lambda], \tag{3h}
$$

$$
\mathbf{x} \in \mathcal{X} \tag{3i}
$$

### 4.3 Data Generation

A diverse dataset of input-output pairs is needed to train Neur2SP's supervised second-stage value approximation. To generate such a dataset for a given 2SP problem, we adopt an iterative procedure. We begin by generating a random feasible first-stage decision. For the NN-E case, we sample a set of scenarios with random cardinality $K'$ from the uncertainty distribution. Here, $K'$ should be chosen to balance the trade-off between the time spent to generate a sample of second-stage values for a given first-stage solution and the time to estimate the expected second-stage value for a set of first-stage decisions in a given time budget. Specifically, if $K'$ is large, then on average more time will be spent in determining the expected value using a large number of scenarios, while for a small $K'$, the first-stage decision space will be explored more since expected value estimates would be

| Problem | First stage | Second Stage | Objective | Constraints | Objective Sense |
|---------|-------------|--------------|-----------|-------------|-----------------|
| CFLP | Binary | Binary | Linear | Linear | Minimization |
| INVP | Continuous | Binary | Linear | Linear | Minimization |
| SSLP | Binary | Binary | Linear | Linear | Minimization |
| PP | Binary | Continuous | Bilinear | Bilinear | Maximization |

Table 1: Problem class characteristics.

obtained faster. For a given input, i.e., a first-stage decision and set of scenarios, we then compute a label by calculating the expected second-stage value $\sum_{k'=1}^{K'} p_{k'} Q_{k'}(\cdot, \boldsymbol{\xi}_{k'})$.

For the NN-P case, at each iteration of the data generation procedure, we sample a single scenario from the uncertainty distribution. For a given input of a first-stage decision and scenario we generate a label by calculating its second-stage value $Q(\cdot, \cdot)$. Last, the input-output pair is added to the dataset.

This data generation procedure is fully parallelizable over the second-stage problems to be solved.

### 4.4 NN-E vs. NN-P Trade-offs

The NN-E and NN-P architectures exhibit trade-offs in terms of the learning task and the resulting surrogate optimization problem.

**Training.**   In data collection, both models require solving second-stage problems with a fixed first-stage solution to obtain the label. A sample in for NN-P requires solving only a single optimization problem, whereas a sample for NN-E requires solving at most $K'$ second-stage problems. As this process is offline and highly parallelizable, this trade-off is easy to mitigate. As for training, NN-E operates on a subset of scenarios which makes for an exponentially larger input space. Despite the large input space, our experiments show that the NN-E model in the training converges quite well and in many cases the embedded model outperforms the NN-P model.

**Surrogate Optimization Problem.**   As the ultimate goal is embedding the trained model into a MIP, the trade-off in this regard becomes quite important. Specifically, for $K$ scenarios, the NN-P model will have $K$ times more binary and continuous variables than the NN-E model. For problems with a large number of scenarios, this makes the NN-E model much more appealing, smaller and likely faster to solve. Furthermore, it allows for much larger networks given that only a single copy of the network is embedded.

## 5   Experimental Setup

All experiments were run on a computing cluster with an Intel Xeon CPU E5-2683 and Nvidia Tesla P100 GPU with 64GB of RAM (for training). Gurobi 9.1.2 [Gurobi Optimization, LLC, 2021] was used as the MIP solver. Scikit-learn 1.0.1 [Pedregosa et al., 2011] and Pytorch 1.10.0 [Paszke et al., 2019] were used for supervised learning. The code to reproduce all of the experiments is available at https://github.com/khalil-research/Neur2SP.

**2SP Problems**: We evaluate our approach on four 2SP problems that are commonly considered in the literature: a two-stage stochastic variant of the Capacitated Facility Location Problem (CFLP) [Cornuéjols et al., 1991], an Investment Problem (INVP) [Schultz et al., 1998], the Stochastic Server Location Problem (SSLP) [Ntaimo and Sen, 2005], and the Pooling Problem (PP) [Audet et al., 2004, Gupte et al., 2017, Gounaris et al., 2009, Haverly, 1978]. Table 1 summarizes the types of first and second-stage variables for these problems and Appendix B includes their detailed descriptions.

**Baselines**: We consider two baselines. The first is EF, which is perhaps the only generic approach that can be applied for both integer and nonlinear second-stage problems. We limit the solving time of EF to 3 hours. Additionally, we compare against an embedded trained linear regressor rather than a neural network, but defer these results to Appendix D as the solution quality is quite poor in comparison to the neural network models.

**Model & Dataset Selection** : As is common in supervised learning, model selection and the size of the training set can have a significant impact on model performance. We present detailed experiments

| Problem | Data Generation Time | | Training Time | | Total Time | |
|---|---|---|---|---|---|---|
| | NN-E | NN-P | NN-E | NN-P | NN-E | NN-P |
| CFLP_10_10 | 1,823.07 | 13.59 | 667.28 | 127.12 | 2,490.35 | 140.71 |
| CFLP_25_25 | 4,148.83 | 112.83 | 2,205.23 | 840.07 | 6,354.06 | 952.90 |
| CFLP_50_50 | 7,697.91 | 135.57 | 463.71 | 128.11 | 8,161.62 | 263.68 |
| SSLP_10_50 | 942.10 | 22.95 | 708.86 | 116.17 | 1,650.96 | 139.13 |
| SSLP_15_45 | 929.27 | 16.35 | 1,377.21 | 229.42 | 2,306.48 | 245.77 |
| SSLP_5_25 | 860.74 | 13.18 | 734.02 | 147.44 | 1,594.75 | 160.62 |
| INVP_B_E | 8,951.27 | 4.17 | 344.87 | 1,000.14 | 9,296.13 | 1,004.31 |
| INVP_B_H | 9,207.90 | 4.22 | 1,214.54 | 607.49 | 10,422.43 | 611.71 |
| INVP_I_E | 8,759.83 | 4.34 | 2,115.25 | 680.93 | 10,875.08 | 685.27 |
| INVP_I_H | 8,944.65 | 3.32 | 393.82 | 174.26 | 9,338.47 | 177.58 |
| PP | 1,202.11 | 14.86 | 576.08 | 367.25 | 1,778.19 | 382.11 |

Table 2: Computing times (in seconds) for data generation and training. Data was generated in parallel with 43 processes.

for model selection and dataset sizing in Appendix F. As a brief summary, we use random search over 100 hyperparameter configurations for model selection, and observe that accuracy on a validation set is rather insensitive to hyperparameter choices. For the size of the dataset, we observe diminishing returns when increasing the size beyond 5000 samples.

**Data Generation & Supervised training times** : As the data generation and training can be done offline and are both parallelizable, we report the total times in Table 2 and defer more specific timing details to Appendix C. We note that for data generation, a single sample can be obtained in less than two seconds for all instances, and in many cases much faster. The training times are within the range of 120 to 2100 seconds. For the NN-E data generation, we choose $K' = 100$, a number of scenarios which was quick to label while exposing the model to a reasonably large number of scenarios in some cases. The combined time for data generation and model training is typically less than 3 hours (i.e., the time given to EF) and depending on the problem, may be much less.

## 6 Results & Discussion

In this section, we report the results of Neur2SP across the four problem settings. As is standard in 2SP, we evaluate a single "base" instance across varying scenario sets and sizes. For example, in CFLP, and for a "base" instance with 10 facilities and 10 customers, one can generate an instance by sampling any number of scenarios. An important advantage of our approach is that we can apply a single trained model to an instance with an arbitrary number of scenarios. For example, the same trained model is used for CFLP_10_10_{100,500,1000}.

Tables 3 through 6 report the gaps between approaches, solving times, and the time which EF takes to achieve the same solution quality as NN-E and NN-P. In addition, we include supplementary results with the objective values in Appendix D and non-aggregated results for the SSLP SIPLib instances in Appendix E. For SSLP and CFLP, each row represents mean statistics across 11 and 10 instances generated by sampling different scenario sets of a given size, respectively. However, for INVP and PP, each row represents the statistics across 1 instance. Originally, both these problems have infinite support as the uncertainty distributions are assumed to be continuous. To manage the complexity, these distributions are typically transformed to have finite support by uniformly sampling equidistant points over the continuous domain. We adopt this same procedure from the literature, leading to a static set of scenarios for a given scenario set size.

### 6.1 Discussion

Tables 3–6 show that NN-E is significantly faster than other approaches, with a minimum and maximum solving time of 0.11s and 1.66s respectively, across all problems. This highlights the scalability of the NN-E in terms of problem size and type, which is expected as the size of the resulting MIP is independent of the number of scenarios. Also, the objective difference is less than 5% in most cases, with a minimum of -102% and a maximum of 13.78%. These differences are inversely proportional to the scenario set size, which indicates that the NN-E is able to generalize on larger scenario sets, even though it was trained with a maximum of 100 scenarios per data point. EF

| Problem | Obj. Difference (%) | | Solving Time | | | EF time to | | | |
|---|---|---|---|---|---|---|---|---|---|
| | EF-NN-E | EF-NN-P | NN-E | NN-P | EF | NN-E | | NN-P | |
| CFLP_10_10_100 | 2.58 | 1.65 | **0.38** | 8.28 | 4,410.60 | 8.87 | (0) | 12.69 | (0) |
| CFLP_10_10_500 | 2.41 | 0.94 | **0.60** | 206.30 | 10,800.17 | 415.89 | (0) | 2,034.73 | (0) |
| CFLP_10_10_1000 | 0.94 | -0.67 | **0.64** | 856.77 | 10,800.87 | 580.50 | (0) | 7,551.00 | (8) |
| CFLP_25_25_100 | -0.75 | -0.75 | **0.44** | 4.86 | 10,800.06 | - | (10) | - | (10) |
| CFLP_25_25_500 | -3.62 | -3.62 | **0.54** | 26.41 | 10,800.14 | - | (10) | - | (10) |
| CFLP_25_25_1000 | -1.32 | -1.32 | **0.58** | 54.45 | 10,800.36 | - | (10) | - | (10) |
| CFLP_50_50_100 | -0.43 | -1.29 | **1.66** | 21.10 | 10,800.05 | 5,637.98 | (6) | 2,334.04 | (9) |
| CFLP_50_50_500 | -9.58 | -10.71 | **1.25** | 173.63 | 10,806.15 | - | (10) | - | (10) |
| CFLP_50_50_1000 | -16.62 | -17.50 | **1.44** | 572.12 | 10,805.82 | - | (10) | - | (10) |

Table 3: CFLP results: each row represents an average over ten 2SP instances with varying scenario sets. "Obj. Difference" for method EF-{NN-E, NN-P} is the percent relative objective value of {NN-E, NN-P} to EF; a negative (positive) value of $-g\%$ ($g\%$) indicates that {NN-E, NN-P} finds a solution that is $g\%$ better (worse) than EF's for the minimization problem. "Solving Time" is the time in which {NN-E, NN-P, EF} are solved to optimality. A time of ∼10,800 implies that the solving limit was reached. "EF time to" is the time in which EF achieves a solution of the same quality as {NN-E, NN-P}. To the right in parentheses is the number of instances for which EF failed to find a solution that is as good as {NN-E, NN-P}. If EF did not find any feasible solution, then the entry is left as "-". All times are in seconds.

| Problem | Obj. Difference (%) | | Solving Time | | | EF time to | | | |
|---|---|---|---|---|---|---|---|---|---|
| | EF-NN-E | EF-NN-P | NN-E | NN-P | EF | NN-E | | NN-P | |
| SSLP_10_50_50 | 0.00 | 0.00 | **0.11** | 5.83 | 10,800.48 | 228.06 | (0) | 228.06 | (0) |
| SSLP_10_50_100 | -0.00 | -0.00 | **0.11** | 13.09 | 10,800.21 | 145.35 | (0) | 145.35 | (0) |
| SSLP_10_50_500 | -0.00 | -0.00 | **0.14** | 129.44 | 10,802.82 | 7,359.85 | (4) | 7,359.85 | (4) |
| SSLP_10_50_1000 | -55.21 | -55.21 | **0.13** | 466.38 | 10,800.47 | - | (11) | - | (11) |
| SSLP_10_50_2000 | -102.69 | -102.69 | **0.14** | 2,182.31 | 10,800.17 | - | (11) | - | (11) |
| SSLP_15_45_5 | 3.10 | 18.71 | **0.32** | 0.34 | 2.54 | 0.75 | (0) | 0.12 | (0) |
| SSLP_15_45_10 | 2.98 | 18.47 | **0.31** | 0.58 | 1,976.62 | 2.72 | (0) | 0.20 | (0) |
| SSLP_15_45_15 | 2.53 | 18.90 | **0.33** | 0.86 | 2,052.76 | 1.84 | (0) | 0.34 | (0) |
| SSLP_5_25_50 | 0.12 | 1.78 | **0.20** | 1.14 | 2.24 | 1.94 | (0) | 1.97 | (0) |
| SSLP_5_25_100 | 0.02 | 1.60 | **0.18** | 1.83 | 8.43 | 8.04 | (0) | 7.75 | (1) |

Table 4: SSLP results: each row represents an average over eleven 2SP instances with varying scenario sets, one of which being the instance from Ahmed et al. [2015]. Columns are as in Table 3.

takes significantly longer to reach a solution quality similar to NN-E, often on the order of minutes to 3 hours. For many larger CFLP and SSLP instances, EF finds worse solutions than NN-E even after 3 hours. Not only is NN-E as good or better in solution quality, but also orders of magnitude faster.

For the NN-P, we can observe that the solving time is directly proportional to the size of the problem. However, for the largest INVP instances, it times out without even generating a feasible solution (Table 5). This is expected as we need to embed the trained neural network once per scenario, limiting scalability. In terms of objective differences, we can observe that the difference improves with the increase in instance size for CFLP and SSLP, whereas no clear trend is visible for For INVP. However, the objective differences do not exceed 5% in most cases. For PP, the objective difference is around 40%, indicating that the NN-P is not able to generalize, whereas NN-E performs very well. One important advantage for NN-P over NN-E is the fact that the time required for data generation and training is notably less. In settings with limited parallel computing resources or time NN-P may be a more appropriate choice.

## 7    Conclusion

Two-stage stochastic programming is a powerful modeling framework for decision-making under uncertainty. These problems are hard to solve in practice, especially when the second-stage problem is a MIP or NLP. Finding good feasible solutions quickly thus becomes extremely important.

To that end, we proposed `Neur2SP`, a learning-based, general, and structure-agnostic approach which approximates the second-stage value function to form an easy-to-solve surrogate problem. The four problem classes we have tackled are (1) widely used in the literature, (2) vary in the types of first and second-stage problems, and (3) span a wide range in terms of number of variables, constraints, and scenarios. Through our experiments, we show that `Neur2SP` achieves high-quality solutions quickly, especially for larger instances. In 1–2 seconds, a model trained in the `Neur2SP` framework can find

| Problem | Obj. Difference (%) | | Solving Time | | | EF time to | |
|---|---|---|---|---|---|---|---|
| | EF-NN-E | EF-NN-P | NN-E | NN-P | EF | NN-E | NN-P |
| INVP_B_E_4 | 9.54 | 3.01 | 0.36 | 0.34 | **0.02** | 0.02 | 0.02 |
| INVP_B_E_9 | 7.54 | 2.00 | 0.31 | 0.53 | **0.04** | 0.03 | 0.03 |
| INVP_B_E_36 | 2.72 | 4.96 | 0.30 | 9.53 | **0.08** | 0.02 | 0.02 |
| INVP_B_E_121 | 1.37 | 2.42 | **0.33** | 86.42 | 1.69 | 0.06 | 0.02 |
| INVP_B_E_441 | 2.80 | 2.43 | **0.37** | 4,342.19 | 117.59 | 0.78 | 1.15 |
| INVP_B_E_1681 | 1.36 | - | **0.34** | - | 10,800.01 | 17.41 | 0.00 |
| INVP_B_E_10000 | -1.48 | - | **0.36** | - | 10,803.98 | - | 0.00 |
| INVP_B_H_4 | 8.81 | 9.50 | 0.46 | 0.25 | **0.01** | 0.01 | 0.01 |
| INVP_B_H_9 | 5.04 | 5.04 | 0.30 | 0.57 | **0.03** | 0.02 | 0.02 |
| INVP_B_H_36 | 1.61 | 1.61 | **0.26** | 6.79 | 1.29 | 0.01 | 0.01 |
| INVP_B_H_121 | 1.77 | 1.77 | **0.33** | 45.89 | 34.69 | 0.01 | 0.01 |
| INVP_B_H_441 | 2.13 | 5.50 | **0.28** | 1,870.42 | 217.46 | 2.21 | 0.21 |
| INVP_B_H_1681 | -0.71 | - | **0.36** | - | 10,800.01 | - | 0.00 |
| INVP_B_H_10000 | -2.72 | - | **0.36** | - | 10,800.03 | - | 0.00 |
| INVP_I_E_4 | 12.83 | 0.00 | 0.38 | 0.23 | **0.01** | 0.01 | 0.01 |
| INVP_I_E_9 | 7.40 | 2.64 | 0.27 | 0.35 | **0.06** | 0.01 | 0.02 |
| INVP_I_E_36 | 5.48 | 5.17 | 0.27 | 1.39 | **0.04** | 0.01 | 0.01 |
| INVP_I_E_121 | 5.30 | 4.49 | **0.29** | 49.51 | 1.65 | 0.02 | 0.03 |
| INVP_I_E_441 | 3.00 | 0.68 | **0.26** | 2,049.93 | 46.92 | 0.08 | 0.10 |
| INVP_I_E_1681 | 1.31 | 3.08 | **0.26** | 10,834.53 | 10,800.00 | 0.41 | 0.41 |
| INVP_I_E_10000 | -1.35 | - | **0.30** | - | 10,800.10 | - | 0.00 |
| INVP_I_H_4 | 13.78 | 12.16 | 0.35 | 0.21 | **0.02** | 0.01 | 0.01 |
| INVP_I_H_9 | 9.12 | 0.81 | 0.37 | 0.31 | **0.03** | 0.01 | 0.02 |
| INVP_I_H_36 | 4.97 | 3.44 | **0.36** | 1.99 | 1.27 | 0.03 | 0.03 |
| INVP_I_H_121 | 4.01 | 4.99 | **0.32** | 23.10 | 7.43 | 0.07 | 0.07 |
| INVP_I_H_441 | 3.15 | 3.15 | **0.32** | 1,231.48 | 10,800.00 | 0.33 | 0.33 |
| INVP_I_H_1681 | -0.34 | 0.11 | **0.33** | 10,816.89 | 10,800.03 | - | 252.70 |
| INVP_I_H_10000 | -1.60 | - | **0.38** | - | 10,802.10 | - | 0.00 |

Table 5: INVP results: each row represents a single instances. Columns are as in Table 3.

| Problem | Obj. Difference (%) | | Solving Time | | | EF time to | |
|---|---|---|---|---|---|---|---|
| | EF-NN-E | EF-NN-P | NN-E | NN-P | EF | NN-E | NN-P |
| PP_125 | 3.25 | 36.64 | **1.51** | 144.08 | 10,800.00 | 10,717.05 | 2.48 |
| PP_216 | 9.06 | 40.14 | **1.47** | 254.94 | 364.98 | 59.79 | 3.59 |
| PP_343 | 0.67 | 40.85 | **1.46** | 570.64 | 10,800.00 | 1,450.54 | 9.12 |
| PP_512 | 8.69 | 39.77 | **1.60** | 1,200.37 | 10,800.01 | 167.77 | 13.80 |
| PP_729 | 1.38 | 37.98 | **1.62** | 3,440.19 | 10,800.01 | 5,867.67 | 36.34 |
| PP_1000 | 5.92 | 41.32 | **1.49** | 10,853.59 | 10,800.00 | 1,596.22 | 210.48 |

Table 6: PP results: each row represents a single instances. Columns are as in Table 3.

solutions of the same or better quality than the most generic method in the literature, EF, with the latter taking minutes to hours.

In terms of future work, this methodology can be extended in many directions. Further innovations in the NN-E model architecture may improve our already positive results. Another direction is the extension of the general idea of embedding trained models into other complex optimization problems, such as bilevel optimization or multi-stage stochastic programming.

Another direction for future work is a more comprehensive comparison of Neur2SP with algorithms that are specialized to a given problem class. However, we note that on SSLP instances, the computing times of progressive hedging [Rockafellar and Wets, 1991], a widely used heuristic for 2SP, is on the order of hours [Torres et al., 2022]. These experiments are not directly comparable as they were run on different hardware. However, this would not meaningfully impact the several order of magnitude reduction in solving time achieved by our approach.

Lastly, in this work, we propose NN-E and NN-P, however, a natural middle ground between these models is a clustering approach which embeds a trained model for a subset of scenarios, rather than a single or the entire scenario set at evaluation time.

**Acknowledgments:** Bodur would like to acknowledge support from an NSERC Discovery Grant. Dumouchelle, Patel, and Khalil acknowledge support from the Scale AI Research Chair Program and an NSERC Discovery Grant.

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
