## A  Symbols

List of symbols used in the paper with their brief description.

| Two-stage stochastic program | |
|---|---|
| $\mathbf{x}$ | First-stage decision vector |
| $\mathbf{c}$ | First-stage objective coefficient vector |
| $K$ | EF scenario set size |
| $k$ | Scenario index |
| $\boldsymbol{\xi}_k$ | $k^{th}$ scenario realization |
| $p_k$ | Probability of scenario $k$ |
| $n$ | Dimension of $\mathbf{x}$ |
| $Q(\mathbf{x}, \boldsymbol{\xi})$ | Second-stage sub-problem for first-stage decision $\mathbf{x}$ and scenario $\boldsymbol{\xi}$ |
| $F(\mathbf{x}, \boldsymbol{\xi})$ | Second-stage cost for first-stage decision $\mathbf{x}$ and scenario $\boldsymbol{\xi}$ |
| $\mathcal{X}$ | Constraint set exclusively on the first-stage decision |
| $\mathcal{Y}(\mathbf{x}, \boldsymbol{\xi})$ | Scenario-specific constraint set for first-stage decision $\mathbf{x}$ and scenario $\boldsymbol{\xi}$ |
| Neural network | |
| $\ell$ | Number of layers in the network |
| $m$ | Index over the neural network layers |
| $d^0$ | Dimensionality of input layer |
| $d^m$ | Dimensionality of layer $m$ |
| $\boldsymbol{\alpha}$ | Input to the neural network |
| $\beta$ | Output of the neural network |
| $W$ | Weight matrix |
| $\mathbf{b}$ | Bias |
| $\sigma$ | Activation function |
| $\mathbf{h}^m$ | $m^{th}$ hidden layer |
| $i$ | Index over the column of weight matrix |
| $j$ | Index over the row of weight matrix |
| $\Phi^1$ | Scenario-encoding network |
| $\Phi^2$ | Post scenario-aggregation network |
| $\Psi^E$ | Scenario-embedding network for NN-E |
| $\Psi^P$ | Scenario-embedding network for NN-P |
| MIP-NN | |
| $\hat{h}$ | Non-negative $ReLU$ input |
| $\check{h}$ | Negative $ReLU$ input |
| $z$ | Indicator variables |
| $\Lambda$ | Number of predictions used in embedding |
| $[M]$ | The set $\{1, \ldots, M\}$ for an $M \in \mathbb{Z}_+$ |

Table 7: Symbols summary

## B  Stochastic Programming Problems

### B.1  Capacitated Facility Location (CFLP)

The CFLP is a decision-making problem in which a set of facility opening decisions must be made in order to meet the demand of a set of customers. Typically this is formulated as a minimization problem, where the amount of customer demand satisfied by each facility cannot exceed its capacity. The two-stage stochastic CFLP arises when facility opening decisions must be made prior to knowing the actual demand. For this problem, we generate instances following the procedure described in [Cornuéjols et al., 1991] and create a stochastic variant by simply generating the first-stage costs and capacities, then generate scenarios by sampling $K$ demand vectors using the distributions defined in [Cornuéjols et al., 1991]). To ensure relatively complete recourse, we introduce additional variables with prohibitively expensive objective costs in the case where customers cannot be served. In the experiments a CFLP with $n$ facilities, $m$ customers, and $s$ scenarios is denoted by CFLP_$n$_$m$_$s$.

## B.2 Investment Problem (INVP)

The INVP is a 2SP problem studied in [Schultz et al., 1998]. This 2SP has a set of continuous first-stage decisions which yield an immediate revenue. In the second stage, after a set of random variables are realized, a set of binary decisions can be made to receive further profit. In this work, we specifically consider the instance described in the example 7.3. of [Schultz et al., 1998]. This problem has 2 continuous variables in the first stage with the domain $[0, 5]$, and 4 binary variables in the second stage. The scenarios are given by two random discrete variables which are defined with equal probability over the range $[5, 15]$. Specifically, for $K$ scenarios, each random variable can take an equally spaced value in the range. Although the number of variables is quite small, the presence of continuous first-stage decision has made this problem relevant within the context of other recent work such as the heuristic approach proposed in [van der Laan and Romeijnders, 2021]. As a note, we reformulate the INVP as an equivalent minimization problem in the remainder of this work. In the experiments an INVP instance is denoted by INVP_$v$_$t$, where $v$ indicates the type of second-stage variable (B for binary and I for integer) and $t$ indicates the type of technology matrix ($E$ for identity and $H$ for $[[2/3, 1/3], [1/3, 2/3]]$).

## B.3 Stochastic Server Location Problem (SSLP)

The SSLP is a 2SP, where in the first stage a set of decisions are made to decide which servers should be utilized and a set of second-stage decisions assigning clients to servers. In this case, the random variables take binary values, which represent a client with a request occurring in the scenario or not. A more detailed description of the problem can be found in [Ntaimo and Sen, 2005]. In this work, we directly use the instances provided in SIPLIB [Ahmed et al., 2015]. In the experiments a SSLP with $n$ servers, $m$ clients, and $s$ scenarios is denoted by SSLP_$n$_$m$_$s$.

## B.4 Pooling Problem (PP)

The pooling problem is a well-studied problem in the field of mixed-integer nonlinear programming Audet et al. [2004], Gupte et al. [2017], Gounaris et al. [2009], Haverly [1978]. It can be formulated as a mixed-integer quadratically constrained quadratic program, making it the hardest problem class in our experiments.

We are given a directed graph, consisting of three disjoint sets of nodes, called the source, pool and terminal nodes. We need to produce and send some products from the source to the terminal nodes, using the given arcs, such that the product demand and quality constraints on the terminal nodes, along with the arc capacity constraints, are satisfied. The pool nodes can be used to mix products with different qualities in appropriate quantities to generate a desired quality product. The goal is to decide the amount of product to send on each arc such that the total profit from the operations is maximized. We consider a stochastic version of the problem as described in the case study of Li et al. [2011]. Here, in the first stage, we need to design the network by selecting nodes and arcs from the input graph, without knowing the quality of the product produced on source nodes and the exact demand on the terminal nodes. Once the uncertainty is revealed, in the second stage, we make the recourse decisions about the amount of product to be sent on each arc, such that demand and quality constraints on the terminal nodes are satisfied. In our case, we have 16 binary variables in the first stage and 11 continuous variables per scenario in the second stage. An instance of this problem is referred to as PP_$s$, where $s$ is the number of scenarios.

# C  Data Generation & Supervised Learning Times

In this section, we report details of the data generation and training times for all problem settings in Tables 8 and 9, respectively. For training, we split the # samples into an 80%-20% train validation set, and select the best model on the validation set in the given number of epochs.

# D  Objective Results

In this section, we report the objective for the first-stage solutions obtained by each approximate MIP and the objective of EF (either optimal or at the end of the solving time). In addition, we

| Problem | NN-E | | | NN-P | | |
|---|---|---|---|---|---|---|
| | # samples | Time per sample | Total time | # samples | Time per sample | Total time |
| CFLP_10_10 | 5,000 | 0.36 | 1,823.07 | 10,000 | 0.00 | 13.59 |
| CFLP_25_25 | 5,000 | 0.83 | 4,148.83 | 10,000 | 0.01 | 112.83 |
| CFLP_50_50 | 5,000 | 1.54 | 7,697.91 | 10,000 | 0.01 | 135.57 |
| SSLP_10_50 | 5,000 | 0.19 | 942.10 | 10,000 | 0.00 | 22.95 |
| SSLP_15_45 | 5,000 | 0.19 | 929.27 | 10,000 | 0.00 | 16.35 |
| SSLP_5_25 | 5,000 | 0.17 | 860.74 | 10,000 | 0.00 | 13.18 |
| INVP_B_E | 5,000 | 1.79 | 8,951.27 | 10,000 | 0.00 | 4.17 |
| INVP_B_H | 5,000 | 1.84 | 9,207.90 | 10,000 | 0.00 | 4.22 |
| INVP_I_E | 5,000 | 1.75 | 8,759.83 | 10,000 | 0.00 | 4.34 |
| INVP_I_H | 5,000 | 1.79 | 8,944.65 | 10,000 | 0.00 | 3.32 |
| PP | 5,000 | 0.24 | 1,202.11 | 10,000 | 0.00 | 14.86 |

Table 8: Data generation samples and times. Data was generated in parallel with 43 processes. All times in seconds.

| | NN-E | NN-P | LR |
|---|---|---|---|
| CFLP_10_10 | 667.28 | 127.12 | 0.53 |
| CFLP_25_25 | 2,205.23 | 840.07 | 0.28 |
| CFLP_50_50 | 463.71 | 128.11 | 0.75 |
| SSLP_10_50 | 708.86 | 116.17 | 0.63 |
| SSLP_15_45 | 1,377.21 | 229.42 | 0.57 |
| SSLP_5_25 | 734.02 | 147.44 | 0.05 |
| INVP_B_E | 344.87 | 1,000.14 | 0.02 |
| INVP_B_H | 1,214.54 | 607.49 | 0.02 |
| INVP_I_E | 2,115.25 | 680.93 | 0.02 |
| INVP_I_H | 393.82 | 174.26 | 0.02 |
| PP | 576.08 | 367.25 | 0.05 |

Table 9: Training times. All times in seconds.

report the objective of the approximate MIP. See Tables 10 through 13 for results. As mentioned in the main paper, the results from linear regressor (LR) are quite poor, with a significantly worse objective in almost every instance. This is not surprising as a linear function will not likely have the capacity to estimate the integer and non-linear second-stage objectives. For both NN-E and NN-P we can see that the true objective and the approximate-MIP objective are relatively close for all of the problem settings, further indicating that the neural network embedding is a useful approximation to the second-stage expected cost.

# E    SSLP SIPLib Results

In this section we report optimally gaps and solving times on the publicly available SSLP SIPLib instances in Table 14. From the table, we can see that both NN-E and NN-P do quite well in terms of finding solutions, especially in the larger scenario case where they obtain optimal first-stage solutions. Perhaps, the most impressive results here is that NN-E is able to obtain optimal results for many instances in ∼0.1 seconds.

# F    Model & Dataset Selection

## F.1    Model Selection

For the supervised learning task, we implement linear regression using Scikit-learn 1.0.1 [Pedregosa et al., 2011]. In this case we use the base estimator with no regularization. The NN-P/NN-E neural models are all implemented using Pytorch 1.10.0 [Paszke et al., 2019]. For model selection, we use random search over 100 configurations for each problem setting. For NN-P and NN-E we sample configurations from Table 15. For both cases we limit the ReLU layers to a single layer with a varying hidden dimension. In the NN-P case the choice of the ReLU hidden dimension is limited since a large

| Problem | True objective | | | | Approximate-MIP objective | | |
|---|---|---|---|---|---|---|---|
| | NN-E | NN-P | LR | EF | NN-E | NN-P | LR |
| CFLP_10_10_100 | 7,174.57 | 7,109.62 | 10,418.87 | **6,994.77** | 7,102.57 | 7,046.37 | 5,631.00 |
| CFLP_10_10_500 | 7,171.79 | 7,068.91 | 10,410.19 | **7,003.30** | 7,102.57 | 7,084.46 | 5,643.68 |
| CFLP_10_10_1000 | 7,154.60 | **7,040.70** | 10,406.08 | 7,088.56 | 7,102.57 | 7,064.36 | 5,622.40 |
| CFLP_25_25_100 | **11,773.01** | **11,773.01** | 23,309.73 | 11,864.83 | 11,811.39 | 12,100.73 | 10,312.21 |
| CFLP_25_25_500 | **11,726.34** | **11,726.34** | 23,310.34 | 12,170.67 | 11,811.39 | 12,051.51 | 10,277.01 |
| CFLP_25_25_1000 | **11,709.90** | **11,709.90** | 23,309.85 | 11,868.04 | 11,811.39 | 12,041.12 | 10,263.37 |
| CFLP_50_50_100 | 25,236.33 | **25,019.64** | 45,788.45 | 25,349.21 | 26,309.43 | 26,004.88 | 18,290.63 |
| CFLP_50_50_500 | 25,281.13 | **24,964.33** | 45,786.97 | 28,037.66 | 26,287.48 | 25,986.50 | 18,209.77 |
| CFLP_50_50_1000 | 25,247.77 | **24,981.70** | 45,787.18 | 30,282.41 | 26,309.43 | 26,002.78 | 18,217.14 |

Table 10: CFLP detailed objective results: each row represents an average over 10 2SP instance with varying scenario sets. "True objective" for {NN-E,NN-P,LR} is the cost of the first-stage solution obtained from the approximate MIP evaluated on the second-stage scenarios. For EF it is the objective at the solving limit. "Approximate-MIP objective" is objective from the approximate MIP for {NN-E,NN-P,LR}. All times in seconds.

| Problem | True objective | | | | Approximate-MIP objective | | |
|---|---|---|---|---|---|---|---|
| | NN-E | NN-P | LR | EF | NN-E | NN-P | LR |
| SSLP_10_50_50 | -354.96 | -354.96 | -63.00 | **-354.96** | -350.96 | -339.42 | -294.69 |
| SSLP_10_50_100 | **-345.86** | **-345.86** | -49.62 | -345.86 | -350.96 | -328.54 | -283.96 |
| SSLP_10_50_500 | **-349.54** | **-349.54** | -54.68 | -349.54 | -350.96 | -332.82 | -288.02 |
| SSLP_10_50_1000 | **-350.07** | **-350.07** | -55.45 | -235.22 | -350.96 | -333.46 | -288.55 |
| SSLP_10_50_2000 | **-350.07** | **-350.07** | -54.72 | -172.73 | -350.96 | -332.87 | -288.19 |
| SSLP_15_45_5 | -247.27 | -206.83 | -249.51 | **-255.55** | -238.44 | -259.11 | -58.28 |
| SSLP_15_45_10 | -249.58 | -209.49 | -252.89 | **-257.41** | -238.44 | -265.92 | -64.01 |
| SSLP_15_45_15 | -251.10 | -208.86 | -254.58 | **-257.68** | -238.44 | -267.01 | -66.71 |
| SSLP_5_25_50 | -125.22 | -123.15 | 14.50 | **-125.36** | -121.64 | -110.18 | -119.98 |
| SSLP_5_25_100 | -120.91 | -119.03 | 19.87 | **-120.94** | -121.64 | -109.59 | -117.79 |

Table 11: SSLP detailed objective results: each row represents an average over eleven 2SP instance with varying scenario sets. See Table 10 for a detailed description of the columns.

number of predictions each with a large hidden dimension can lead to MILPs which are prohibitively expensive to solve. For the NN-E specific hidden dimensions, we have 3 layers, with Embed hidden dimension 1 and Embed hidden dimension 2 corresponding to layers before the aggregation and Embed hidden dimension 3 being a final hidden layer after the aggregation.

In Tables 16 and 17 we report the best parameters for each problem setting for the NN-P and NN-E models, respectively. In addition, we report the validation MSE across all 100 configurations for each problem in box plots in Figures 3 to 6. From the box plots we can observe that lower validation MAE configurations are quite common as the medians are typically not too far from the lower tails of the distributions. This indicates that hyperparameter selection can be helpful when attempting to improve the second-stage cost estimates, however, the gains are marginal in most cases.

## F.2 Dataset Size Selection

In this section, we report results for varying dataset sizes. Here, we report only the results for a single problem setting, namely, CFLP_10_10. We use a validation with 5000 samples and training sets with 100, 500, 1000, 5000, 10000, and 20000 samples. Model selection with random search is done for each training set size as described in the previous section. Figures 7 and 8 report the results for the NN-P and NN-E models respectively. In both cases, we can see an improvement in validation MAE with increases in the dataset sizes, however, diminishing returns start to occur when increasing the number of samples above 5000 samples. This motivates the choice of dataset sizes which we use in the remainder of the experiments. Specifically, we use 10000 samples for the NN-P case as data generation is quite fast. For the NN-E case we limit the number of samples to 5000 as we only see a small improvement of %4 in validation MAE at the cost of doubling the compute time.

| Problem | True objective | | | | Approximate-MIP objective | | |
|---|---|---|---|---|---|---|---|
| | NN-E | NN-P | LR | EF | NN-E | NN-P | LR |
| INVP_B_E_4 | -51.56 | -55.29 | -46.25 | **-57.00** | -58.59 | -52.15 | -63.67 |
| INVP_B_E_9 | -54.86 | -58.15 | -53.11 | **-59.33** | -58.81 | -55.33 | -63.67 |
| INVP_B_E_36 | -59.55 | -58.19 | -58.86 | **-61.22** | -59.38 | -57.92 | -63.67 |
| INVP_B_E_121 | -61.44 | -60.78 | -61.06 | **-62.29** | -59.60 | -58.91 | -63.67 |
| INVP_B_E_441 | -59.60 | -59.83 | -59.91 | **-61.32** | -59.91 | -58.51 | -63.67 |
| INVP_B_E_1681 | -59.81 | - | -59.30 | **-60.63** | -59.94 | - | -63.67 |
| INVP_B_E_10000 | **-59.85** | - | -58.68 | -58.98 | -59.95 | - | -63.67 |
| INVP_B_H_4 | -51.75 | -51.36 | -51.75 | **-56.75** | -58.12 | -52.41 | -61.24 |
| INVP_B_H_9 | -56.56 | -56.56 | -56.56 | **-59.56** | -61.78 | -56.67 | -61.24 |
| INVP_B_H_36 | -59.31 | -59.31 | -59.31 | **-60.28** | -59.38 | -59.52 | -61.24 |
| INVP_B_H_121 | -59.93 | -59.93 | -59.93 | **-61.01** | -60.22 | -60.54 | -61.24 |
| INVP_B_H_441 | -60.14 | -58.07 | -60.14 | **-61.44** | -60.23 | -58.13 | -61.24 |
| INVP_B_H_1681 | **-60.47** | - | **-60.47** | -60.04 | -60.57 | - | -61.24 |
| INVP_B_H_10000 | **-60.53** | - | **-60.53** | -58.93 | -60.65 | - | -61.24 |
| INVP_I_E_4 | -55.35 | **-63.50** | -52.50 | **-63.50** | -66.79 | -58.96 | -71.57 |
| INVP_I_E_9 | -61.63 | -64.80 | -61.89 | **-66.56** | -66.70 | -61.70 | -71.57 |
| INVP_I_E_36 | -66.03 | -66.25 | -67.08 | **-69.86** | -67.39 | -65.18 | -71.57 |
| INVP_I_E_121 | -67.35 | -67.92 | -69.07 | **-71.12** | -67.39 | -66.70 | -71.57 |
| INVP_I_E_441 | -67.55 | -69.16 | -67.39 | **-69.64** | -67.63 | -67.43 | -71.57 |
| INVP_I_E_1681 | -67.95 | -66.73 | -66.52 | **-68.85** | -67.69 | -67.62 | -71.57 |
| INVP_I_E_10000 | **-67.94** | - | -65.67 | -67.04 | -67.82 | - | -71.57 |
| INVP_I_H_4 | -54.75 | -55.78 | -54.75 | **-63.50** | -65.31 | -59.99 | -66.07 |
| INVP_I_H_9 | -59.78 | -65.25 | -59.78 | **-65.78** | -64.15 | -61.08 | -66.07 |
| INVP_I_H_36 | -63.78 | -64.80 | -63.78 | **-67.11** | -66.79 | -63.76 | -66.07 |
| INVP_I_H_121 | -65.03 | -64.37 | -65.03 | **-67.75** | -65.38 | -64.64 | -66.07 |
| INVP_I_H_441 | -65.12 | -65.12 | -65.12 | **-67.24** | -67.13 | -65.16 | -66.07 |
| INVP_I_H_1681 | **-65.63** | -65.34 | **-65.63** | -65.41 | -65.87 | -65.03 | -66.07 |
| INVP_I_H_10000 | **-65.66** | - | **-65.66** | -64.63 | -66.45 | - | -66.07 |

Table 12: INVP detailed objective results: each row represents single instance. See Table 10 for a detailed description of the columns.

| Problem | True objective | | | | Approximate-MIP objective | | |
|---|---|---|---|---|---|---|---|
| | NN-E | NN-P | LR | EF | NN-E | NN-P | LR |
| PP_125 | 264.30 | 173.10 | -10.00 | **273.19** | 189.75 | 177.12 | 70.75 |
| PP_216 | 200.29 | 131.83 | -10.00 | **220.25** | 189.75 | 168.10 | 70.75 |
| PP_343 | 206.38 | 122.90 | -10.00 | **207.77** | 189.75 | 172.17 | 70.75 |
| PP_512 | 204.41 | 134.83 | -10.00 | **223.86** | 189.75 | 162.54 | 70.75 |
| PP_729 | 219.42 | 137.97 | -10.00 | **222.48** | 189.75 | 167.55 | 70.75 |
| PP_1000 | 202.50 | 126.30 | -10.00 | **215.25** | 189.75 | 165.59 | 70.75 |

Table 13: PP detailed objective results: each row represents single instance. See Table 10 for a detailed description of the columns.

| Problem | Gap to Optimal (%) | | | Solving Time | | |
|---|---|---|---|---|---|---|
| | NN-E | NN-P | EF | NN-E | NN-P | EF |
| SSLP_10_50_50 | **0.00** | **0.00** | **0.00** | **0.11** | 4.83 | 10,801.27 |
| SSLP_10_50_100 | **0.00** | **0.00** | **0.00** | **0.11** | 11.66 | 10,800.04 |
| SSLP_10_50_500 | **0.00** | **0.00** | 0.00 | **0.11** | 107.88 | 10,818.23 |
| SSLP_10_50_1000 | **0.00** | **0.00** | 28.64 | **0.12** | 383.51 | 10,800.26 |
| SSLP_10_50_2000 | **0.00** | **0.00** | 51.24 | **0.13** | 4,523.19 | 10,800.20 |
| SSLP_15_45_5 | 0.46 | 19.59 | **0.00** | 0.32 | **0.28** | 4.17 |
| SSLP_15_45_10 | 1.57 | 18.23 | **0.00** | 0.25 | 0.40 | 3.71 |
| SSLP_15_45_15 | 0.53 | 16.51 | **0.00** | 0.41 | 0.72 | 4.74 |
| SSLP_5_25_50 | **0.00** | 2.15 | **0.00** | 0.26 | 0.92 | 2.35 |
| SSLP_5_25_100 | **0.00** | 1.40 | **0.00** | 0.18 | 1.68 | 8.87 |

Table 14: SSLP SIPLib gap and time comparison among methods. Optimal SIPLib instances values obtained from Ahmed et al. [2015]. "Gap to Optimal" is the percent gap to the optimal solution. "Solving Time" is the solving to of the approximate MIP and EF. All times in seconds.

| Parameter | NN-P | NN-E |
|---|---|---|
| Batch size | $\{16, 32, 64, 128\}$ | $\{16, 32, 64, 128\}$ |
| Learning rate | $[1e^{-5}, 1e^{-1}]$ | $[1e^{-5}, 1e^{-1}]$ |
| L1 weight penalty | $[1e^{-5}, 1e^{-1}]$ | $[1e^{-5}, 1e^{-1}]$ |
| L2 weight penalty | $[1e^{-5}, 1e^{-1}]$ | $[1e^{-5}, 1e^{-1}]$ |
| Optimizer | {Adam, Adagrad, RMSprop} | {Adam, Adagrad, RMSprop} |
| Dropout | $[0, 0.5]$ | $[0, 0.5]$ |
| # Epochs | 1000 | 2000 |
| ReLU hidden dimension | $\{32, 64\}$ | $\{64, 128, 256, 512\}$ |
| Embed hidden dimension 1 | - | $\{64, 128, 256, 512\}$ |
| Embed hidden dimension 2 | - | $\{16, 32, 64, 128\}$ |
| Embed hidden dimension 3 | - | $\{8, 16, 32, 64\}$ |

Table 15: Random search parameter space for NN-P and NN-E models. For values in { }, we sample with equal probability for each discrete choice. For values in [], we sample a uniform distribution with the given bounds. For single values, we keep it fixed across all configurations. A value of - means that parameter is not applicable for the given model type.

| Parameter | CFLP_10_10 | CFLP_25_25 | CFLP_50_50 | SSLP_5_25 | SSLP_10_50 | SSLP_15_45 | INVP_B_E | INVP_B_H | INVP_I_E | INVP_I_H | PP |
|---|---|---|---|---|---|---|---|---|---|---|---|
| Batch size | 128 | 16 | 128 | 128 | 128 | 64 | 16 | 32 | 32 | 128 | 64 |
| Learning rate | 0.05029 | 0.05217 | 0.00441 | 0.03385 | 0.07015 | 0.08996 | 0.00435 | 0.00521 | 0.06613 | 0.01614 | 0.0032 |
| L1 weight penalty | 0.07512 | 0.00551 | 0.08945 | 0.03232 | 0.07079 | 0.09105 | 0.08321 | 0.05754 | 0.01683 | 0.01859 | 0 |
| L2 weight penalty | 0.08343 | 0.02846 | 0.08602 | 0.0 | 0.01792 | 0.0 | 0.01047 | 0.02728 | 0 | 0 | 0.0361 |
| Optimizer | Adam | Adam | Adam | RMSprop | RMSprop | RMSprop | RMSProp | RMSProp | Adam | Adam | Adam |
| Dropout | 0.02198 | 0.02259 | 0.05565 | 0.00914 | 0.01944 | 0.02257 | 0.17237 | 0.13698 | 0.04986 | 0.0859 | 0.05945 |
| ReLU hidden dimension | 64 | 32 | 64 | 32 | 64 | 32 | 64 | 64 | 64 | 32 | 64 |

Table 16: NN-P best configurations from random search.

| Parameter | CFLP_10_10 | CFLP_25_25 | CFLP_50_50 | SSLP_5_25 | SSLP_10_50 | SSLP_15_45 | INVP_B_E | INVP_B_H | INVP_I_E | INVP_I_H | PP |
|---|---|---|---|---|---|---|---|---|---|---|---|
| Batch size | 32 | 16 | 128 | 64 | 64 | 32 | 128 | 32 | 16 | 128 | 64 |
| Learning rate | 0.0263 | 0.06571 | 0.02906 | 0.08876 | 0.07633 | 0.03115 | 0.01959 | 0.00846 | 0.02841 | 0.02867 | 0.08039 |
| L1 weight penalty | 0.02272 | 0.06841 | 0.05792 | 0.0 | 0.04529 | 0.07182 | 0.0 | 0.00022 | 0 | 0 | 0 |
| L2 weight penalty | 0.05747 | 0.0 | 0.04176 | 0.03488 | 0.0 | 0.0 | 0 | 0.09007 | 0.02272 | 0.01882 | 0 |
| Optimizer | RMSprop | Adam | Adam | Adam | RMSprop | Adam | Adagrad | Adam | Adagrad | Adagrad | Adam |
| Dropout | 0.01686 | 0.0028 | 0.03318 | 0.00587 | 0.00018 | 0.0088 | 0.08692 | 0.04096 | 0.01854 | 0.01422 | 0.0072 |
| ReLU hidden dimension | 128 | 256 | 256 | 256 | 64 | 256 | 256 | 256 | 256 | 256 | 512 |
| Embed hidden dimension 1 | 512 | 128 | 256 | 64 | 128 | 512 | 256 | 512 | 64 | 256 | 512 |
| Embed hidden dimension 2 | 16 | 64 | 64 | 16 | 32 | 64 | 16 | 16 | 32 | 32 | 128 |
| Embed hidden dimension 3 | 16 | 16 | 8 | 32 | 64 | 16 | 32 | 16 | 8 | 64 | 16 |

Table 17: NN-E best configurations from random search.

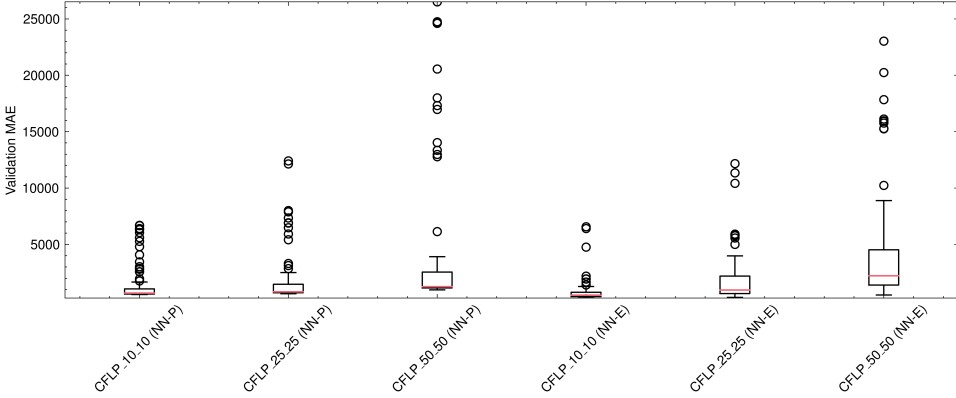

Figure 3: CFLP validation MAE over random search configurations for NN-P and NN-E models.

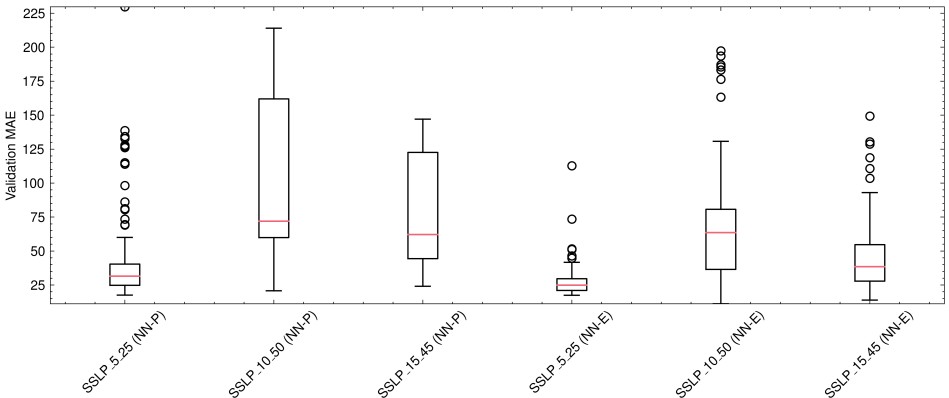

Figure 4: SSLP validation MAE over random search configurations for NN-P and NN-E models.

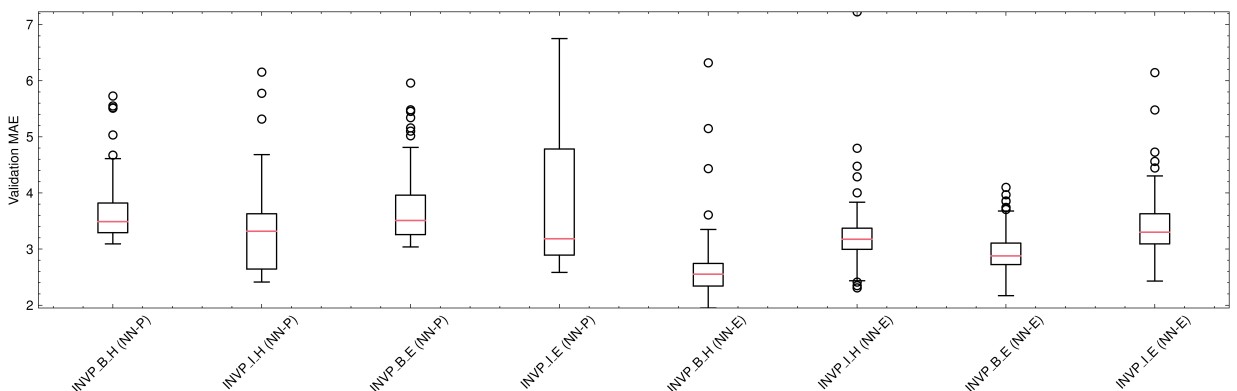

Figure 5: INVP validation MAE over random search configurations for NN-P and NN-E models.

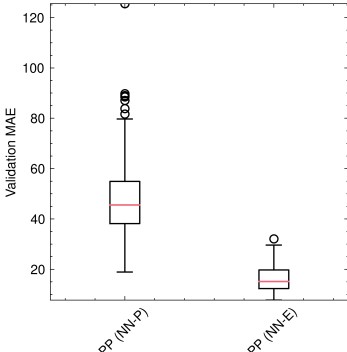

Figure 6: PP validation MAE over random search configurations for NN-P and NN-E models.

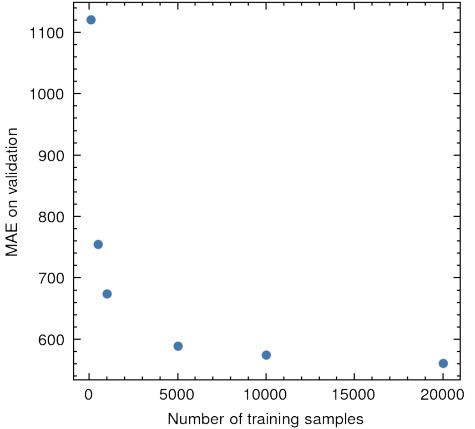

Figure 7: NN-P dataset sizing results

Figure 8: NN-E dataset sizing results