# OpenReview forum: "Neur2SP: Neural Two-Stage Stochastic Programming"
_NeurIPS.cc/2022/Conference — NeurIPS 2022 Accept_

### Official Review · Reviewer_FXNT · 2022-07-11

**Rating:** 7
**Confidence:** 4
**Soundness:** 3 good
**Presentation:** 3 good
**Contribution:** 4 excellent

**Summary:**

This paper presents an algorithm for producing feasible solutions for two-stage stochastic optimization problems. The main approach is to train a neural network to approximate the second-stage value function, and then embed this neural network into the first stage optimization problem. The authors present a deep learning architecture for learning the second-stage value function in both a "single-cut" and "multi-cut" setting, and discuss the data generation scheme for supervised learning. The computational section shows that the new "single-cut" method produces solutions that are typically roughly as good, and sometimes superior to, the "extended formulation", and arrives there in far less time (seconds vs. hours).



**Questions:**

* Have the authors considered how their method would generalize to problems with incomplete recourse? There is a glancing mention about artificially inducing relatively complete recourse in the appendix, but no other discussion. At the very least, please add a brief discussion about any assumptions on Y in the Preliminaries section, and a bit of discussion on the resulting limitations.
* The "Multi-cut" subsection in 4.1 seems oddly abbreviated: is there nothing of interest to say about the architecture used? The equation block (3) also seems out of place.
* How are valid(?) bounds computed for the input of the value function? This is a crucial ingredient necessary to determine the validity and computational performance of the end-to-end MIP formulation. In particular, my understanding from the paper is that the downstream MIP formulation only ever considers the latent space learned to represent the aggregated scenarios. How are bounds computed in this space?
* I am curious if the authors have done any sensitivity analysis regarding the randomness inherent in the NN training procedure. More specificially, is there a wide variance in downstream performance for models trained with differing random seeds, for example? As intuitively it seems that the MIP optimizer will attempt to identify and take advantage of any artificial locally optimal "artifact" in the model, I am curious if randomness affects the presence or location of these artifacts, and if this ends up affecting the downstream performance.
* In the computational section, can the authors provide more distributional information about the objective difference and solve time (and also, potentially, the number of "time outs")? I am especially curious about Table 5, and if the relatively poor performance of EF-MC w.r.t. objective difference can be attributed primarily to instances where it "timed out", or if the performance is just uniformly worse across the board.

**Limitations:**

The authors do not discuss the potential negative societal impact of their work.

**Strengths And Weaknesses:**

The paper is clearly written, and presents an interesting and seemingly novel approach to a problem of great interest to the operations research community. The computational results suggest that the approach can offer a compelling tradeoff between solution quality and solve time.

---

> ### Author Response · Authors · 2022-08-02
> **Response to reviewer FXNT**
>
>
>
> 1. In the stochastic programming (SP) community, it is a standard assumption to have a relatively complete recourse; in fact, many stochastic programs arising from specific applications indeed satisfy a stronger version of this property, namely, the complete recourse condition [1]. In case it is not readily satisfied, it can be easily ensured by introducing slack variables to (a subset of) the second-stage constraints along with a penalty to the objective function, as such incurring a high cost for an infeasible second-stage solution. This assumption is actually very useful in practice [2]. Therefore, in the SP community, this is indeed deemed as a good modeling practice, thus not considered a limiting assumption. The reason is that infeasibility is technically almost always unavoidable in practice since there would be a chance of observing realizations of uncertain parameters outside of the support of the empirical probability distribution. However, there is always a feasible recourse action in practice, despite highly undesirable outcomes.
>
> 	Nevertheless, if one would like to work with an SP model that does not have relatively complete recourse, then one can add feasibility cuts, both during the data generation time and in an iterative fashion during test time to handle infeasibility by refining the first-stage feasible set.
>
> 2. The multi-cut model is simply a feed-forward network that takes in a single scenario and a first-stage decision vector and predicts the second-stage cost as described in subsection 4.1.  As the architecture is quite simple we do not discuss it much, however, in the revision, we can make an explicit note on the architecture.
>
>
> 3. Neur2SP is a data-driven heuristic solution technique. Hence, theoretically, we cannot obtain bounds using it. However, empirically we can observe that the optimal objective of the surrogate MIP is indeed close to the actual objective when evaluated on the true scenarios using the first-stage solution from the surrogate (See Appendix D for details). We note that a learning theoretic analysis would be a useful result for providing bounds, however, this is more so a direction for future work.
>
> 4. We have not done any analysis regarding the impact of seed on the downstream task. However, we did a random search for different hyperparameter settings, which had low variance on the validation error (See Appendix F). Since we have the optimal hyperparameter configuration, we can include such an analysis in the appendix. However, we do not believe that the downstream results would drastically vary as the variance in the predictive models was quite small.
>
> 5. We will add the number of timeouts in the tables. In addition, we will include distributional information for the objective and solving time. For Table 5, we have a single instance per row as the scenarios are sampled in a structured manner. Hence, there is only one instance which timed out for the multi-cut approach, specifically PP_1000. Also, looking at the results for other scenario sizes, the multi-cut approach performs poorly for the Pooling problem.
>
> [1] Rockafellar, R. Tyrrell, and R. JB Wets. "Stochastic convex programming: relatively complete recourse and induced feasibility." SIAM Journal on Control and Optimization 14.3 (1976): 574-589.
>
> [2] Birge, John R., and Francois Louveaux. Introduction to stochastic programming. Springer Science & Business Media, 2011.

---

> > ### Comment · Reviewer_FXNT · 2022-08-08
> > **Clarification**
> >
> > >> How are valid(?) bounds computed for the input of the value function? This is a crucial ingredient necessary to determine the validity and computational performance of the end-to-end MIP formulation. In particular, my understanding from the paper is that the downstream MIP formulation only ever considers the latent space learned to represent the aggregated scenarios. How are bounds computed in this space?
> >
> > > Neur2SP is a data-driven heuristic solution technique. Hence, theoretically, we cannot obtain bounds using it. However, empirically we can observe that the optimal objective of the surrogate MIP is indeed close to the actual objective when evaluated on the true scenarios using the first-stage solution from the surrogate (See Appendix D for details). We note that a learning theoretic analysis would be a useful result for providing bounds, however, this is more so a direction for future work.
> >
> > My original question was a bit underspecified, so let me clarify: I am referring to variable bounds in the MIP model on the inputs to the value function, not optimality bounds on the end SP problem. These input variable bounds can (should) be used to determine the magnitude of the "big-M" coefficients used to formulate the nonlinear ReLU activations in the MIP model. Can the authors please clarify how these big-M coefficients are computed in their implementation? I am particularly interested because, while in the original space it may be relatively straightforward to compute "reasonable" bounds on the inputs by inspection, after the translation to the latent space I imagine most/all interpretability is lost.

---

> > > ### Author Response · Authors · 2022-08-08
> > > **Response to Clarification**
> > >
> > > Thank you for the clarification.  We use a sufficiently large big-M value of 100,000 based on preliminary experiments.  Specifically, we note that the pre-activation value for the ReLU hidden units are on the order of +/-100s for both MC and SC.  As such the big-M value of 100,000 will be valid for the instances we consider.
> > >
> > > We acknowledge that choosing the large big-M is obviously not ideal as it increases the solving time of our approach; a smaller big-M can only yield tighter LP relaxations.  We note some better methods for obtaining better valid bounds are the following:
> > > - Allowing Gurobi to handle the bounds with logical constraints (https://www.gurobi.com/documentation/9.5/refman/py_model_agc_indicator.html)
> > > - Once we have a trained ReLU network, the weights can be leveraged to obtain bounds by solving an optimization problem over the first stage decision variables to determine the lowest valid big-M for a given scenario or set of scenarios.  As either the scenario or set of scenarios are fixed during evaluation (i.e. solving a surrogate MIP), the bounds computed by big-M with respect to either a scenario for multi-cut or the latent representation can be computed similary.  We believe this specifically addresses the last point of your comment.
> > > - Lastly, the most suitable approach would be to leverage existing techniques for choosing big-M values in ReLU MIPs, e.g. [1,2].
> > >
> > > [1] Serra, Thiago, and Srikumar Ramalingam. "Empirical bounds on linear regions of deep rectifier networks." Proceedings of the AAAI Conference on Artificial Intelligence. Vol. 34. No. 04. 2020.
> > >
> > > [2] Grimstad, Bjarne, and Henrik Andersson. "ReLU networks as surrogate models in mixed-integer linear programs." Computers & Chemical Engineering 131 (2019): 106580.

---

### Official Review · Reviewer_D2XQ · 2022-07-11

**Rating:** 8
**Confidence:** 4
**Soundness:** 4 excellent
**Presentation:** 4 excellent
**Contribution:** 4 excellent

**Summary:**

The authors propose a method for quickly obtaining high quality solutions to two stage stochastic programs (2sp) by training a neural network to estimate the second stage solution quality given the first stage solution, and then solving for the first stage solution by encoding the second stage estimator in the first stage MIP formulation, replacing the second stage value with it’s estimator. The authors evaluate on a variety of realistic 2sp instances with both linear and bilinear second stage problems demonstrating the flexibility of their approach to a variety of problems. Overall the authors demonstrate orders of magnitude of performance improvement over reasonable baselines that are equivalently general, making the class of 2sp problems much more accessible for realistic problems.

**Questions:**

Is it possible to compare against embedding other predictive models in the MIP formulation?

Are there any settings in which the method didn't work, or instance in settings where the first or second stage is highly combinatorial?

Is this approach applicable for bilevel optimization settings? What would be the potential pitfalls there?

**Limitations:**

The authors adequately addressed limitations and potential negative social impact

**Strengths And Weaknesses:**

The core strength of the paper is in its contribution to making a general class of 2sp problems computationally tractable in a sensible manner, combining the expressive power of neural networks with the computational power of MIP solvers to obtain high-quality solutions to 2sp problems quickly. The paper is very clear, original, and significant.

Additionally, the experiments and motivation are sound making this a great contribution to the literature of improving optimization performance with machine learning.

I think the main area that could be improved is to investigate the sample efficiency of the approach and potential for other predictive models such as linear models or decision trees which could also be encoded as MIP. Otherwise, the approach seems to be sound and have great significance for the optimization community or anyone who is tackling problems of two stage stochastic programs. It would also be interesting to see whether this could be employed for other bilevel optimization tasks such as those that arise in game theory where the second level task is difficult to solve. Finally, it would be interesting to see if the same approach could work for more stages of decision making.

---

> ### Author Response · Authors · 2022-08-02
> **Response to reviewer D2XQ**
>
> 1. Yes, other predictors can be embedded easily for the multi-cut approach.  We include a comparison for an embedded linear model in the Tables 9-12 in Appendix D where we show significantly better performance.  For the single-cut case the neural network was a design choice as it allows one to obtain the compact scenario representation in an end-to-end learning procedure.  The extension of this to other predictors is less clear as we rely on the permutation invariant model to embed the scenarios end-to-end.
> 2. It worked in all of the settings we tried so far without any difficulty. We are actively looking for more challenging instances to further test our method on.
> 3. In principle, the general idea for embedding a predictor in a MIP is likely possible in the bilevel optimization case, but it may not be straightforward. Note that the standard bilevel programs aims to find x and y optimizing F(x,y), where x is an upper-level feasible solution whereas y is a lower-level optimal solution given x. Therefore, in order to get rid of y and end up with a monolithic formulation, we would need the representation of the optimality conditions for the lower-level problem. Having an approximation of the lower-level objective function may not be enough for that aim.

---

### Official Review · Reviewer_fXmS · 2022-07-12

**Rating:** 7
**Confidence:** 4
**Soundness:** 4 excellent
**Presentation:** 4 excellent
**Contribution:** 4 excellent

**Summary:**

The paper introduces a heuristic method for two-stage stochastic optimization problems that approximates the value function of the second stage with a neural network. Notably, the problem may be mixed-integer, possibly nonlinear, in the second stage, which can make the problem particularly challenging. The value function for a problem is learned in advance by sampling scenarios and training a neural network, which is embedded into the first-stage problem as a MIP. Computational results on a variety of stochastic optimization instances show positive results compared to the extensive form of the problem.

**Questions:**

No questions or suggestions besides the above. A minor detail is that Equation (3) should be in section 4.2.

**Limitations:**

The paper should be more explicit regarding data generation and training time as discussed above, but there are no limitations on societal impact that need to be addressed.

**Strengths And Weaknesses:**

This paper provides an effective heuristic to solve two-stage stochastic optimization problems, which has several important applications in Operations Research. An appealing property of this method is its generality, as it can be applied without any modifications even when the second stage is mixed-integer or nonlinear, which often makes the problem more challenging. The computational experiments are reasonably extensive and positive: the method can produce better results than the extensive form for certain large-scale instances, particularly when the number of scenarios is large, and sometimes even when taking into account data generation and training time sequentially. There are also instances where the method does not perform as well when taking training time into account (INVP), which is useful to have in the paper. The paper is overall well-written and clear.

My concerns with the paper are the following:

1. The text is not very explicit when taking into account the data generation and training time, often implicitly treating as if they were not significant. Granted, they can be parallelized and the learned function can generalize to a larger number of scenarios, but that does not mean they can be dismissed, particularly because as is, this approximation is not intended to generalize to other instances. In particular, I do not believe it is fair to present the solution time in the abstract without proper contextualization (i.e. that this is without data generation and training time), and Section 6.1 should also be framed within this context. Ideally, I would prefer to see training and data generation times in the main text and I would encourage the authors to find space to fit them in, but I understand that this is difficult due to space issues. Another side effect of this is that, as is, at first glance the multicut approach does not seem very useful, but if you take into account data generation and training time it becomes more appealing particularly if one does not have access to broad parallelization capabilities, and perhaps this can be pointed out in Section 6.1.

2. One limitation is that the paper does not compare with decomposition algorithms for two-stage stochastic optimization problems (e.g. Benders-based methods, progressive hedging). Decomposition methods tend to be substantially better than the extensive form in practice, and thus this paper does not answer the question of whether one should choose this approach over a decomposition method. It is also worth noting that this method is a heuristic and the extensive form is exact, which is not an ideal comparison (although I do appreciate the column "EF time to" in the tables). On the other hand, an argument is that the method presented in the paper is more general. In any case, although this is a limitation that should not be ignored, in my opinion this is outweighed by the rest of the paper.

Given that item 1 above is addressed, in my opinion this paper should be accepted. Not only this approach might be immediately applicable in practice for an important class of optimization problems (although not too clear due to the lack of comparison with decomposition methods), it can also be a useful stepping stone for further papers applying similar ideas. Therefore, I believe this paper can have a high impact in the field.

---

> ### Author Response · Authors · 2022-08-02
> **Response to reviewer fXms**
>
> 1. We agree with the observation that having the computation time in the main text would be beneficial for the overall readability of the paper. We would try to rearrange and fit the training times in the main text. Also, we would add the point about the usability of the multi-cut approach in the absence of parallel computation.
> 2. We agree that having more baselines, progressive hedging and decomposition methods, would be useful. As you mention, our approach is more general, which was the primary motivation for only comparing to another general approach (EF).  We note that applying progressive hedging, i.e., iteratively updating the anticipative solutions and Lagrangian multipliers, is possible. However, there may not be convergence guarantees and the solved problems would be computationally expensive for the non-linear case.   For other decomposition techniques, we note that as the variable domains vary between instances it will require more specific decomposition implementations.  For instance, Benders decomposition can be only applied to the linear problems with purely continuous second-stage variables, the integer L-shaped method requires only binary first-stage variables to be linked to the second-stage problem, and their most general version called the logic-based Benders decomposition necessitates the design of problem-specific cuts.

---

> > ### Comment · Reviewer_fXmS · 2022-08-09
> > **Rebuttal acknowledgement**
> >
> > Thank you for the response. I appreciate that the camera-ready version would be more upfront about the training and data collection time, given that this is not the typical case where you would generalize for a wide set of instances.
> >
> > I understand the reasons not to include stronger baselines. This is however in my opinion the main drawback of the paper: even though decomposition approaches may not have the same generality as this approach, it would have been very useful to know whether one may prefer this approach over traditional decomposition approaches, which are most commonly used. That said, I believe that the current contributions outweigh this issue and my hope is that this paper sparks future work that can answer this question properly.

---

### Official Review · Reviewer_9pYt · 2022-07-26

**Rating:** 4
**Confidence:** 4
**Soundness:** 2 fair
**Presentation:** 3 good
**Contribution:** 2 fair

**Summary:**

This paper studies two-stage stochastic programs (2SPs). To address the computational complexity associated with the second-stage problem, the key idea is to use a neural network to approximate the second-stage value that facilitates the first-stage optimization. Both single-cut and multi-cut approximations are considered. Finally, Neur2SP is validated via experiments.

**Questions:**

- The key idea of using a neural network to approximate the otherwise computationally-demanding second-stage problem has been well explored in a number of prior studies (not cited), so the novelty of this paper is rather limited. For example, the following papers use neural networks to approximate the solution of an inner maximization/minimization problem in worst-case robust optimization. While the specific contexts are a bit different from the one in this paper, the main technical novelty (i.e., using a neural network to speed up the inner optimization) in those papers still apply here.

[1] Learning A Minimax Optimizer: A Pilot Study, ICLR'21.
[2] Learning for Robust Combinatorial Optimization: Algorithm and Application, INFOCOM'22.
[3] Learning to Defend by Learning to Attack, AISTATS'21.
[4] Improved Adversarial Training via Learned Optimizer, ECCV'20.

- The training distribution for the neural network to approximate the second-stage optimization is pre-generated/-determined, but it's entangled with the first-stage solution. As a result, the actual testing distribution (depending on the first-stage solution, which itself also depends on the second-stage approximation) can differ from the pre-generated distribution. This can raise convergence and/or out-of-distribution issues.

- The single-cut vs. multi-cut tradeoff is only briefly touched without an in-depth analysis.

- Secton 2.2 (Embedding Neural Networks into MIPs) is interesting, but this is taken directly from a prior study.

**Strengths And Weaknesses:**

Strengths:
+ The paper is fairly well written and easy to follow.
+ 2SP potentially has practical applications.

Weakness:
- The key idea of using a neural network to approximate the otherwise computationally-demanding second-stage problem has been well explored in a number of prior studies (not cited), so the novelty of this paper is rather limited. For example, the following papers use neural networks to approximate the solution of an inner maximization/minimization problem in worst-case robust optimization. While the specific contexts are a bit different from the one in this paper, the main technical novelty (i.e., using a neural network to speed up the inner optimization) in those papers still apply here.

[1] Learning A Minimax Optimizer: A Pilot Study, ICLR'21.
[2] Learning for Robust Combinatorial Optimization: Algorithm and Application, INFOCOM'22.
[3] Learning to Defend by Learning to Attack, AISTATS'21.
[4] Improved Adversarial Training via Learned Optimizer, ECCV'20.

- The training distribution for the neural network to approximate the second-stage optimization is pre-generated/-determined, but it's entangled with the first-stage solution. As a result, the actual testing distribution (depending on the first-stage solution, which itself also depends on the second-stage approximation) can differ from the pre-generated distribution. This can raise convergence and/or out-of-distribution issues.

- The single-cut vs. multi-cut tradeoff is only briefly touched without an in-depth analysis.

- Secton 2.2 (Embedding Neural Networks into MIPs) is interesting, but this is taken directly from a prior study.

---

> ### Author Response · Authors · 2022-08-02
> **Response to reviewer 9pYt**
>
> 1. We acknowledge the fact that there exists literature for predictions with respect to inner optimization problems and we will add them accordingly to the related work. We realize that Neur2SP can be classified in a similar manner, however, there are a few important differences since Neur2SP was designed for two-stage discrete problems. Specifically,
>
> 	- All the above-cited methods focus on predicting the solution directly using neural networks. This is easily achieved as the output is either continuous or binary.  Whereas Neur2SP integrates a trained prediction model within a classical optimization technique, a MIP in this case, to directly handle the variable domains in addition to any hard constraints on the outer optimization problem.
>
>
> 	- As a result of the above point, none of the cited references are designed to be used in the case of hard (and even non-linear) constraints for the inner or outer problems, outside the scope of limited variable domains (i.e., binary or continuous).  Whereas Neur2SP can be straightforwardly applied.
>
> 	- Reference [2] is closest to the problem class we consider as they have addressed the case of binary decision variables. In [2], only the outer problem is binary, whereas Neur2SP can be applied when both the inner and outer problems are binary (and more generally integer). In addition, the computational results are limited to only one relatively small robust optimization problem in [2], whereas Neur2SP is evaluated on significantly larger benchmark instances for four different two-stage stochastic programming problems. Lastly, the methodology in [2] is designed for the robust case, whereas we focus on the expectation over inner optimization problems. For this reason, the extension of [2] to the stochastic case is non-trivial.
>
> 	- References [1,3,4] all deal with continuous unconstrained bi-level problems. As the optimization problem we are dealing with is discrete, the use of any of the methods proposed in these papers is not obvious, and perhaps not possible.
>
> 	That being said, we agree that these should all be included in the related work.
>
> 2. In the training phase, Neur2SP is learning to predict the second-stage cost with respect to an input given by a first-stage decision and scenario pair.  In this case the first-stage decisions are sampled randomly to train the network. In the test phase, the exploration over the first-stage decision space is explored through the MIP, while the second-stage cost is approximated with the embedded trained neural network. We do acknowledge that there may be out-of-distribution issues as we only train on a subset of first-stage decision scenario pairs, however, the strong empirical results demonstrate that out-of-distribution is not an issue for the problems we consider.
> 3. In section 4.4 we include a discussion of two of the more important differences between the single- and multi-cut approaches, namely, the learning and downstream trade-offs between the models. Specifically, for data generation the multi-cut approach requires less computation to produce a single sample. However, for downstream MIP complexity, the multi-cut model requires on the order of K times the number of decision variables compared to the single-cut approach.
> 4. We acknowledge that this section pertains to a previous study that was cited. This section was included in the background as it is an important component of how we formulate the surrogate optimization model for two-stage problems. However, we note that the use of MIP embeddings of neural networks in the context of stochastic programming is new, and enables a principled optimization over the trained value function approximation model. We believe these to be strong contributions of our work.

---

> > ### Comment · Reviewer_9pYt · 2022-08-07
> > **Thanks for the response.**
> >
> > I appreciate the authors' response, but my biggest concern --- the novelty of using a neural network to approximate the inner optimization --- still remains. Some prior studies, including the ones I referenced in my earlier review, may predict the optimal solution to the inner problem, whereas Neur2SP predicts the optimal value of the inner problem. This is not a key differentiator. In fact, predicting the optimal solution (typically a multi-dimensional vector) is even more challenging and general than predicting a single optimal value. With the predicted optimal solution, one can trivially obtain the optimal value by plugging the predicted solution into the objective function, but not vice versa. Whether the outer problem is continuous, binary, or mixed-integer programming, has nearly no impact on the use of a neural network to approximate either the solution to or the value of the inner problem.
> >
> > I stand by my point given that the use of a neural network to approximate the inner problem is the key *novelty and contribution* of this paper but has been well explored in the literature. More specifically, I do not view the empirical application of a well-explored idea to a variant problem setting, without new rigorous theoretical analysis or significant modification to the already-explored method, as a major novelty or contribution.
> >
> > PS: By "application of a well-explored idea", I meant the use of a neural network to approximate the inner optimization problem.

---

> > > ### Author Response · Authors · 2022-08-08
> > > **Thank you and a response**
> > >
> > > Thank you for engaging with our rebuttal! We appreciate the opportunity to further address your key concern regarding novelty.
> > >
> > > __1. Positioning our work:__ We do not claim that approximating the inner optimization problems’ objective function is our key contribution. In fact, it is not, and we will build on your constructive review and the references you’ve kindly shared to emphasize that point in the next iteration of the paper. Our contribution is in designing an ML-based heuristic algorithm for two-stage stochastic programming problems. Achieving this ambitious goal has required not only function approximation, but also careful neural network architecture design to accommodate an extremely large number of scenarios (the single-cut architecture) and also an integration of the trained model in a final MILP that produces the heuristic solutions.
> > >
> > >
> > > __2. The importance of our “application”:__ Two-stage stochastic programming is a modeling framework rather than an application. As such, Neur2SP enables the solution of not one real application, but a very wide range of problems that can be modeled in the two-stage SP language. Stochastic Programming is a very large subfield of Mathematical Programming, with even a conference of its own (https://na.eventscloud.com/website/40825/), and a large community of researchers; for instance, the OptimizationOnline pre-print repository (the arXiv of Math. Optimization), counts 919 Stochastic Programming papers, one of the biggest categories: https://optimization-online.org/repository/. We have tackled a diverse set of 4 applications in our paper, something which is extremely rare in the SP literature due to the specialized algorithm design philosophy. That we achieve near-optimal solutions in a few seconds for all instances of these problems using the single-cut architecture is extremely promising with wide-reaching implications for this field.
> > >
> > >
> > > __3. Predicting the value vs. the solution:__ Yes, the latter seems more challenging at face value, but it is in fact not even possible in our setting. Given that we have hard constraints in both stages, it is extremely unlikely that any model will be able to output a feasible solution directly. This is where embedding the trained model into a final “heuristic” MILP becomes crucial. Papers [1–4] all treat unconstrained problems (min-max/robust) and thus may be amenable to the prediction of a solution directly. However, as we will show in the next point 4, they can only do so in restricted settings and/or at very small scale.
> > >
> > >
> > > __4. Scale and generality of the problems being tackled:__
> > > In terms of problem scale as measured by the number of decision variables, some of our instances have __millions__ of variables, for example the CFLP_50_50_1000 instances have 50 first-stage variables and 2.5 million second-stage variables, and the SSLP_10_50_2000 instances have 10 first-stage variables and 1 million second-stage variables.
> > >
> > > In contrast, in the paper [1] that you’ve kindly referenced, the min-max problem at hand is unconstrained, and the number of decision variables in the experiments is tiny, 2 for Seesaw and Rotated Saddle, and 25 for Matrix Game, a limitation that is acknowledged in the Conlusion of that paper and its title (“a pilot study”).
> > >
> > > In [2], again, an unconstrained optimization problem is addressed with only a single experiment that involves only 20 decision variables.
> > >
> > > In [3] and [4], the important adversarial training problem is addressed. While of wide interest in the deep learning literature, it is only one (also unconstrained) problem.
> > >
> > > Our work, while addressing a different class of problems than papers [1–4], is both widely applicable to many two-stage stochastic programming problems and simultaneously scales to extremely large numbers of variables. We believe that this makes our contributions here at least comparable to, if not more substantial, than [1–4].

---

### Comment · Area_Chair_YGeF · 2022-08-06
**Please make sure the authors' response has been read**

Dear Reviewers,

Thanks for providing the review. The discussion stage will end in next Tuesday. Please check the authors' response and feel free to discuss with authors.

Best,
AC

---

### Meta-Review · Area_Chair_YGeF · 2022-08-26

**Recommendation:** Accept
**Confidence:** Certain

**Metareview:**

In this paper, the authors proposed to use learning with neural network to amortize the cost in the two-stage optimization problems. The authors tested the algorithms on several problems, demonstrating the advantages of the proposed algorithm empirically.

Most of the reviewers think this work is interesting, although there are already plenty of existing work considering the similar methods, especially a similar work has been published that using learning to amortize for multi-stage stochastic programming.

Another concern raised by reviewer is that the empirical comparison is not comprehensive. The decomposition methods, e.g., Benders-based methods, are not involved, which is a major algorithm for two-stage stochastic optimization problems. Therefore, the advantages of the proposed method is not clear.

Please take the reviewers' points into account to improve the paper.

**Award:**

No

---

### Decision · Program_Chairs · 2022-09-14

Accept